# Adaptive Data-Knowledge Alignment in Genetic Perturbation Prediction

**Yuanfang Xiang**
Nanjing University
yf.xiang@smail.nju.edu.cn

**Lun Ai**
EMBL-EBI
ail@ebi.ac.uk

## Abstract

The transcriptional response to genetic perturbation reveals fundamental insights into complex cellular systems. While current approaches have made progress in predicting genetic perturbation responses, they provide limited biological understanding and cannot systematically refine existing knowledge. Overcoming these limitations requires an end-to-end integration of data-driven learning and existing knowledge. However, this integration is challenging due to inconsistencies between data and knowledge bases, such as noise, misannotation, and incompleteness. To address this challenge, we propose ALIGNED (Adaptive aLignment for Inconsistent Genetic kNowledgE and Data), a neuro-symbolic framework based on the Abductive Learning (ABL) paradigm. This end-to-end framework aligns neural and symbolic components and performs systematic knowledge refinement. We introduce a balanced consistency metric to evaluate the predictions' consistency against both data and knowledge. Our results show that ALIGNED outperforms state-of-the-art methods by achieving the highest balanced consistency, while also re-discovering biologically meaningful knowledge. Our work advances beyond existing methods to enable both the transparency and the evolution of mechanistic biological understanding.

## 1 Introduction

Understanding how genetic perturbation affects transcriptional regulation is essential for deciphering complex biological systems, with profound implications for drug discovery and precision medicine (Badia-i Mompel et al., 2023; Gavriilidis et al., 2024; Ahlmann-Eltze et al., 2025). While advances in experimental technology now allow systematic interrogation of gene regulatory landscapes at an unprecedented scale (Norman et al., 2019; Replogle et al., 2022), existing datasets remain insufficient for building predictive models that can elucidate the full complexity of a cellular system (Peidli et al., 2024). This raises a critical question of how to design predictive frameworks that not only achieve high accuracy but also yield deeper biological understanding from these experimental capabilities.

Two complementary approaches have emerged, either by leveraging latent representations trained on extensive cell data (Lotfollahi et al., 2023; Theodoris et al., 2023; Cui et al., 2024; Hao et al., 2024) or incorporating prior biological knowledge for inductive biases (Roohani et al., 2024; Wang et al., 2024; Littman et al., 2025; Wenkel et al., 2025). Yet, both approaches provide limited insights into the biological mechanisms underlying their predictions. Data-driven models operate as black boxes, making it difficult to understand which regulatory relationships drive specific predictions (Bendidi et al., 2024). While hybrid methods incorporate prior biological knowledge, they treat this knowledge as static constraints rather than interpretable and updatable representations of biological understanding. Importantly, current approaches provide no end-to-end solution to identify and resolve divergences between data-driven learning and existing knowledge, which limits opportunities for continual refinement of biological understanding (Gavriilidis et al., 2024; Zuidberg Dos Martires et al., 2024; Kedzierska et al., 2025).

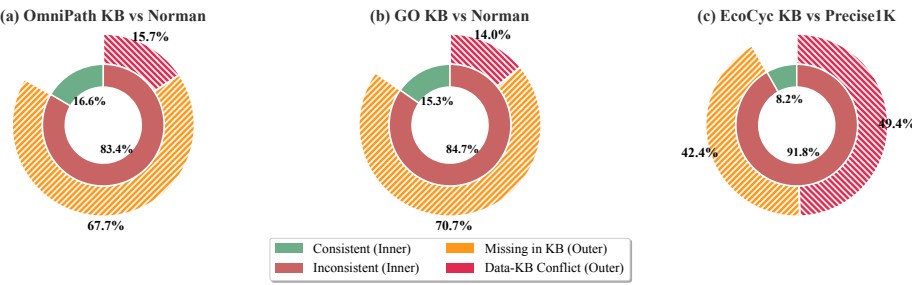

Figure 1: Inconsistency between gene regulatory knowledge bases (KBs) and data-derived perturbation-responses correlations. We examined OmniPath (Türei et al., 2016), Gene Ontology (GO) (Ashburner et al., 2000) and EcoCyc (Moore et al., 2024) knowledge bases, human (Norman et al., 2019) and bacterial (Precise1k, Lamoureux et al., 2023) datasets.

Overcoming these limitations requires explicitly integrating data-driven learning with established knowledge. However, a key challenge is the pervasive inconsistencies between experimental data and curated knowledge (Lu et al., 2024) due to imperfections in both information sources. Perturbation datasets exhibit multiple sources of noise (Liu et al., 2025; Rohatgi et al., 2024), experimental measurement biases (Kim et al., 2015; Peidli et al., 2024) and weak post-perturbation signals (Nadig et al., 2025; Aguirre et al., 2025). Meanwhile, transcriptional regulatory knowledge bases curated by experts often suffer from outdated information (Khatri et al., 2012), limited coverage (Saint-André, 2021) and biases towards better-studied pathways (Chevalley et al., 2025).

To illustrate this challenge, we analyzed popular knowledge bases and benchmark datasets (Figure 1), finding that 42-71% of data-derived regulatory relationships are missing across curated knowledge bases, while a minimum of 14% directly conflict with existing annotations. Naive integration of inconsistent sources risks bidirectional error propagation (Lu et al., 2024) that can corrupt both data-driven learning and knowledge refinement. This inconsistency prevents models from effectively leveraging prior biological knowledge in predictions (Ahlmann-Eltze et al., 2025) and compromises their ability to produce biologically meaningful regulatory relationships from learned representations.

To address this challenge, the Abductive Learning (ABL) paradigm (Zhou, 2019; Huang et al., 2023) offers a foundation for integrating data-driven learning with symbolic knowledge refinement through consistency optimization. Based on this approach, we propose ALIGNED (Adaptive aLignment for Inconsistent Genetic kNowledgE and Data), an end-to-end framework that enables neuro-symbolic alignment and knowledge refinement in genetic perturbation prediction. ALIGNED advances beyond existing predictive methods to enhance transparency about the underlying biological mechanisms and enable continual evolution of understanding from large-scale perturbation datasets.

Our main contributions are:

- **Balanced Consistency Metric.** We design a balanced evaluation metric that assesses predictions against both experimental data and curated knowledge. This addresses the limitation that standard metrics evaluate only predictive accuracy without considering consistency with biological knowledge (Bendidi et al., 2024).

- **Adaptive Neuro-Symbolic Alignment.** We adaptively combine neural and symbolic predictions from inconsistent information sources by weighting neural and symbolic components with a gradient-free optimization mechanism.

- **Knowledge Refinement.** We enable systematic update of regulatory interactions by introducing a gradient-based optimization approach over a symbolic representation of the GRNs.

- **Results**. ALIGNED matches or exceeds state-of-the-art methods in predictive accuracy while substantially outperforms existing methods in balanced consistency. In addition, we show that ALIGNED's knowledge refinement can re-discover cross-referenced regulatory relationships.

## 2 Preliminaries

### 2.1 Problem Setting

We formalize the prediction of genome-scale response to genetic perturbation as a ternary classification problem. The goal is to learn a function $f : \{-1, 0, 1\}^n \to \{-1, 0, 1\}^n$, where $n$ is the total number of genes. The input values $-1$, $0$, and $1$ represent negative perturbation (deletion or knockout), no perturbation, and positive perturbation (overexpression), respectively. The output values indicate decreased expression, no significant change, or increased expression for each gene. Ternary representation of gene expression profiles is commonly used in downstream biological research tasks, such as gene differential-expression analysis and pathway annotations (Love et al., 2014; Badia-i Mompel et al., 2023).

We denote the labelled dataset by $D_l = \langle \boldsymbol{X}_l, \boldsymbol{Y}_l \rangle$, where $\boldsymbol{X}_l$ contains the perturbed gene inputs and $\boldsymbol{Y}_l$ contains the corresponding perturbation responses obtained from transcriptome sequencing experiments. An unlabelled dataset $\boldsymbol{X}_u$ is also used for training in abductive learning, which contains only the perturbation input.

### 2.2 Symbolic reasoning over gene regulatory networks

We focus on gene regulatory networks (GRNs) as our knowledge bases, which contain activation (+) and inhibition (-) interaction relations between genes. It is a widely used qualitative approach to capture directed regulatory effects (Chevalley et al., 2025). We utilize symbolic reasoning via Boolean matrices (Ioannidis & Wong, 1991; Ai, 2025). Direct activation and inhibition interactions are compiled as $n \times n$ adjacency matrices $\langle \boldsymbol{R}_+^{(0)}, \boldsymbol{R}_-^{(0)} \rangle$:

$$\boldsymbol{R}_+^{(i)} = \boldsymbol{R}_+^{(0)} \cdot \boldsymbol{R}_+^{(i-1)} + \boldsymbol{R}_-^{(0)} \cdot \boldsymbol{R}_-^{(i-1)}$$
$$\boldsymbol{R}_-^{(i)} = \boldsymbol{R}_+^{(0)} \cdot \boldsymbol{R}_-^{(i-1)} + \boldsymbol{R}_-^{(0)} \cdot \boldsymbol{R}_+^{(i-1)} \tag{1}$$

We approximate the fixpoint of $\langle \boldsymbol{R}_+^{(\infty)}, \boldsymbol{R}_-^{(\infty)} \rangle$ by interleaving the computations with respect to a partial ordering on the matrices $\boldsymbol{R}_+^{(k)}, \boldsymbol{R}_-^{(k)}$ for a finite $k$ (Tarski, 1955). The obtained knowledge base $\mathcal{KB} = \langle \boldsymbol{R}_+^{(k)}, \boldsymbol{R}_-^{(k)} \rangle$ represents indirect regulations via pathways up to a maximum length of $k$ interactions.

Given an input perturbation $\boldsymbol{x}$, we infer its effect on a genome scale by performing a deductive query in the knowledge base $\mathcal{KB}$. The matrix operations $\delta_{\mathcal{KB}}(\boldsymbol{x}) = (\boldsymbol{R}_+^{(k)} - \boldsymbol{R}_-^{(k)})^\top \boldsymbol{x}$ allow us to perform this query with high computational efficiency. Based on this approach, we define a measurement for the data-knowledge inconsistency illustrated in Figure 1:

$$\text{Inc}(D_l, \mathcal{KB}) = \sum_{\boldsymbol{x}, \boldsymbol{y} \in D_l} \|\delta_{\mathcal{KB}}(\boldsymbol{x}) - \boldsymbol{y}\|_0 \tag{2}$$

where $D_l = \langle \boldsymbol{X}_l, \boldsymbol{Y}_l \rangle$ is a labelled dataset. Our approach differs from the Known Relationships Retrieval metric (Celik et al., 2024; Bendidi et al., 2024) in that the deductive queries respect the global GRN structure and preserve the transitivity of genetic interactions.

### 2.3 Abductive Learning

Our framework explicitly integrates the neural and symbolic components and handles data-knowledge inconsistencies based on the Abductive learning (ABL) paradigm (Zhou, 2019). ABL is a neuro-symbolic approach that aims to learn a function $f$ and align its predictions with the knowledge base $\mathcal{KB}$ via consistency optimization.

A general ABL training pipeline takes a neural model $f$ pretrained on labelled data $\langle \boldsymbol{X}_l, \boldsymbol{Y}_l \rangle$ as initialization. From the unlabelled dataset $\boldsymbol{X}_u$, $f$ makes neural predictions $\hat{\boldsymbol{y}} = f(\boldsymbol{x}_u)$, which may be inconsistent with $\mathcal{KB}$. Consistency optimization is then performed, with revising $\hat{\boldsymbol{y}}$ to $\bar{\boldsymbol{y}}$ and updating $f$ on the revised dataset $\langle \boldsymbol{X}_u, \bar{\boldsymbol{Y}} \rangle$. This process can be executed iteratively until convergence or reaching an iteration limit $T$. Formally, the objective of ABL

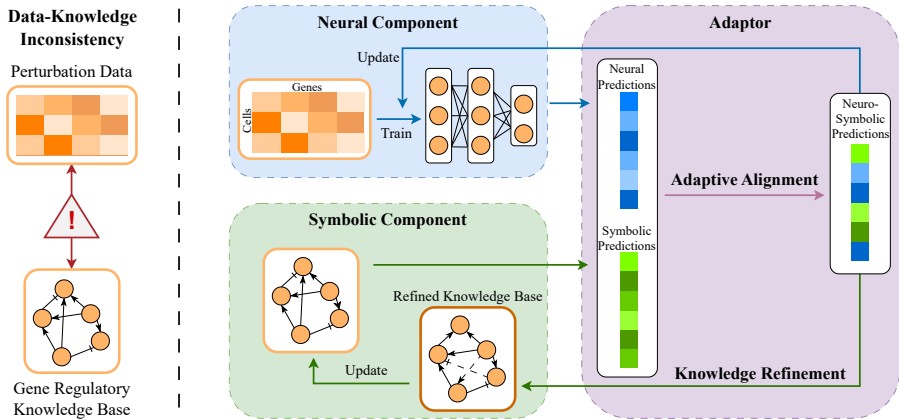

Figure 2: The ALIGNED (Adaptive aLignment of Inconsistent Genetic kNowledgE and Data) framework. ALIGNED contains a neural component (blue), a symbolic component (green) and an adaptor (purple).

is to make predictions that are most consistent with $\mathcal{KB}$:

$$\max_{f}\ p(\boldsymbol{y} \mid \boldsymbol{x}; f, \mathcal{KB}) = \mathbb{I}\big(\mathcal{KB} \cup \boldsymbol{x} \models f(\boldsymbol{x})\big)\ p(\boldsymbol{y} \mid \boldsymbol{x}; f)$$

where on the right hand side "$\models$" denotes logical entailment, the first term is an indicator of consistency and the second term denotes the posterior distribution of neural predictions.

## 3 THE ALIGNED METHOD

We first introduce a consistency metric for evaluating how well predictions align with both the test data and the knowledge base. We then present ALIGNED (Adaptive aLignment for Inconsistent Genetic kNowledgE and Data), a framework which adaptively integrates reliable information from both sources to predict genetic perturbation responses.

### 3.1 THE BALANCED CONSISTENCY METRIC

To evaluate the consistency of a prediction against both the test data and the knowledge base, we define a balanced consistency metric $F_{1\ \text{balance}}$, which considers the macro $F_1$ scores from both the test dataset and the knowledge base. $F_{1\ \text{balance}}$ includes a coefficient $\gamma > 1$ to balance the two macro $F_1$ scores and penalize when either score being too low:

$$F_{1\ \text{balance}}(f(\boldsymbol{x}), \boldsymbol{x}, \boldsymbol{y}, \mathcal{KB}) = \big(\frac{1}{2}F_1(\boldsymbol{y}, f(\boldsymbol{x}))^{-\gamma} + \frac{1}{2}F_1(\delta_{\mathcal{KB}}(\boldsymbol{x}), f(\boldsymbol{x}))^{-\gamma}\big)^{-1/\gamma} \tag{3}$$

This metric evaluates the model's performance on making generalizable predictions while remaining grounded in curated biological prior. Unless specified, all $F_1$ scores in evaluations are computed as macro $F_1$.

### 3.2 ALIGNED FRAMEWORK OVERVIEW

The ALIGNED framework (Figure 2) integrates three components to leverage reliable information from data and knowledge for predicting perturbation response and refining regulatory knowledge. The neural component $f_y$ is a neural network which predicts perturbation responses from input data, while the symbolic component $\mathcal{KB}$ performs symbolic reasoning over gene regulatory networks encoded as matrices (computed via Equation 1). The adaptor $f_a$ learns to combine neural and symbolic predictions based on their relative reliability for each prediction. ALIGNED assumes that (i) when perturbation data is relatively more reliable than prior knowledge, the adaptor would increase reliance on the neural component,

(ii) when the prior knowledge provides more reliable information, the symbolic component would contribute more to overall predictions.

Training proceeds using both labelled and unlabelled data. We initialize components $f_y$ and $f_a$ by training them jointly on the labelled dataset. For each unlabelled input, the framework produces a neural and a symbolic prediction. In adaptive alignment, since these predictions may be inconsistent, the adaptor is trained to produce a binary indicator vector that selects which predictive source to trust for each output dimension. This creates an integrated neuro-symbolic prediction that combines results from both predictive sources. The framework then performs multiple iterations of alignment and bidirectional updates to neural and symbolic components. Using the neural-symbolic predictions, we re-train the neural component and perform knowledge refinement to the symbolic component.

### 3.3 Adaptive Neuro-Symbolic Alignment with Gradient-Free Optimization

In this section, we will introduce the alignment mechanism used by ALIGNED to adaptively integrate neural and symbolic predictions. We denote a binary alignment indicator vector $\boldsymbol{a} = f_a(\boldsymbol{x})$ and the neuro-symbolic prediction $\bar{\boldsymbol{y}}$. After the initialization and each round of bidirectional update, $\bar{\boldsymbol{y}}$ is produced from both neural prediction $\hat{\boldsymbol{y}}$ and symbolic prediction $\delta_{\mathcal{KB}}(\boldsymbol{x})$ according to the indicator $\boldsymbol{a}$, such that $\bar{y}_i = \begin{cases} \hat{y}_i, & a_i = 0 \\ \delta_{\mathcal{KB}}(\boldsymbol{x})_i, & a_i = 1 \end{cases}$.

Our definition of the training objectives for the adaptor can be divided into three parts. First, since information derived from experimental data and curated knowledge may be inconsistent, the adaptor considers how neuro-symbolic predictions differ from the curated knowledge. We describe this using the inconsistency between $\bar{\boldsymbol{y}}$ and $\mathcal{KB}$ based on Equation 2:

$$\text{Inc}(\boldsymbol{a}, \boldsymbol{x}, \hat{\boldsymbol{y}}, \mathcal{KB}) = \|\delta_{\mathcal{KB}}(\boldsymbol{x}) - \bar{\boldsymbol{y}}\|_0$$

Second, we design a loss term to leverage as much information from labelled training data as possible to reduce the predictions' inconsistency with data. Therefore, we restrict the framework to only use knowledge-derived information when necessary. We defined the knowledge usage restriction with threshold $\theta$:

$$L_{len}(\boldsymbol{a}) = \max\{\|\boldsymbol{a}\|_0 - \theta, 0\}$$

Third, we take into account how well each gene is represented in knowledge. To measure this, we use a weight vector $\boldsymbol{w}$ as hyperparameter, which contains the number of training data samples that are inconsistent with $\mathcal{KB}$ (computed by Equation 2), and the number of annotations from Gene Ontology (Ashburner et al., 2000). This allows us to reward $\boldsymbol{a}$ by maximizing the usage of symbolic prediction when a gene is represented well in $\mathcal{KB}$, otherwise use neural prediction. We define a loss with regard to $\boldsymbol{w}$:

$$L_{weight}(\boldsymbol{a}) = \boldsymbol{w}^\top(\mathbf{1} - \boldsymbol{a}) + (\mathbf{1} - \boldsymbol{w})^\top \boldsymbol{a}$$

where higher values of $w_i$ indicate that gene $i$ is well-represented in the knowledge base and more consistent with data, suggesting the symbolic prediction should be preferred. Lower values indicate sparser knowledge or more data-knowledge conflicts, and so the neural prediction should be favored. We combine the above three parts in the adaptor's objective:

$$L_a(\boldsymbol{a}, \boldsymbol{x}, \hat{\boldsymbol{y}}) = \text{Inc}(\boldsymbol{a}, \boldsymbol{x}, \hat{\boldsymbol{y}}, \mathcal{KB}) + C_l L_{len}(\boldsymbol{a}) + C_w L_{weight}(\boldsymbol{a}) \quad (4)$$

where hyperparameter $C_w$, $C_l$ are trade-off coefficients. Minimizing $L_a$ includes querying the symbolic $\mathcal{KB}$, which has a discrete structure. This creates a combinatorial optimization problem, so a gradient-free optimization method is necessary. We train $f_a$ with the REINFORCE algorithm (Williams, 1992; Hu et al., 2025) and initialize its sampling distribution based on $\boldsymbol{w}$ to reduce sampling complexity. To exploit representations captured by the neural component $f_y$ from the experimental data, $f_a$ shares input $\boldsymbol{x}$ and embedding layers with $f_y$. **Adaptive alignment** optimizes $f_y$ and $f_a$ jointly with the following objective:

$$\min_{f_y, f_a} \mathcal{L} = \frac{1}{|D_l|} \sum_{(\boldsymbol{x}, \boldsymbol{y}) \in D_l} CE(f_y(\boldsymbol{x}), \boldsymbol{y})$$

$$+ C \frac{1}{|D_l \cup D_u|} \sum_{\boldsymbol{x} \in D_l \cup D_u} L_a(\boldsymbol{a}, \boldsymbol{x}, \hat{\boldsymbol{y}}) \log f_a(\boldsymbol{x}) \quad (5)$$

where $L_a(\boldsymbol{a}, \boldsymbol{x}, \hat{\boldsymbol{y}})$ does not involve gradient passing. $CE(\cdot, \cdot)$ denotes the cross-entropy loss function, $C$ is a trade-off coefficient, $D_l$ and $D_u$ are labelled and unlabelled datasets.

## 3.4 Gradient-Based Knowledge Refinement with Sparse Regularization

To address missing and inaccurate interactions in $\mathcal{KB}$, we incorporate a knowledge refinement mechanism into the ALIGNED framework that leverages reliable information from neural and symbolic predictions. For computational efficiency on large-scale GRNs, we consider gradient-based optimization, and introduce an approximation function $\varepsilon(\cdot)$ for Boolean elements (Ravanbakhsh et al., 2016). This approximation enables gradient-based optimization compatibility of the non-differentiable Boolean matrix operations in Equation 1:

$$\varepsilon_t(\boldsymbol{X})_{i,j} = 1 - \exp(-tX_{i,j}), X_{i,j} \geq 0$$

We introduce an inductive bias for minimal modifications to the GRN during refinement. This ensures the biological relationships and structure in the GRN are not distorted by spurious signals from data, such as overfitting to noise or learning shortcut patterns. We perform an $l_1$ sparse regularized optimization to achieve this, fitting $\mathcal{KB}$ to neuro-symbolic predictions using proximal gradient descent (Tibshirani, 1996; Candes & Recht, 2012). **Knowledge refinement** optimizes the following objective:

$$\min_{\bar{\boldsymbol{R}}_+^{(0)}, \bar{\boldsymbol{R}}_-^{(0)} \in \mathbb{R}_+^{n \times n}} \mathcal{L}_{\text{refine}}(\bar{\boldsymbol{R}}_+^{(0)}, \bar{\boldsymbol{R}}_-^{(0)}, k) = \sum_{\boldsymbol{x}, \boldsymbol{y} \in \langle \boldsymbol{X}_u, \bar{\boldsymbol{Y}} \rangle} \| \varepsilon_{t_k}(\bar{\boldsymbol{R}}_+^{(k)} - \bar{\boldsymbol{R}}_-^{(k)})^\top \boldsymbol{x} - \boldsymbol{y} \|_2^2$$

$$+ \lambda \big( \| \varepsilon_{t_0}(\bar{\boldsymbol{R}}_+^{(0)}) - \boldsymbol{R}_+^{(0)} \|_1 + \| \varepsilon_{t_0}(\bar{\boldsymbol{R}}_-^{(0)}) - \boldsymbol{R}_-^{(0)} \|_1 \big) \quad (6)$$

where $\mathcal{KB}$ is substituted with $\langle \varepsilon_{t_k}(\bar{\boldsymbol{R}}_+^{(k)}), \varepsilon_{t_k}(\bar{\boldsymbol{R}}_-^{(k)}) \rangle$ after the knowledge refinement step.

$\boldsymbol{R}_+^{(0)}$ and $\boldsymbol{R}_-^{(0)}$ represent the initial GRN before refinement, real-valued non-negative matrices $\bar{\boldsymbol{R}}_+^{(0)}$ and $\bar{\boldsymbol{R}}_-^{(0)}$ are the refined GRN with direct regulatory interactions. To facilitate gradient passing, we use real-number matrix computation instead of Boolean matrix in Equation 1 and use $\bar{\boldsymbol{R}}_+^{(0)}$ and $\bar{\boldsymbol{R}}_-^{(0)}$ to compute indirect regulatory interactions $\bar{\boldsymbol{R}}_+^{(k)}$ and $\bar{\boldsymbol{R}}_-^{(k)}$. $\lambda$ denotes a regularization parameter, $t_k, t_0$ denote coefficients of approximation, $\| \cdot \|_1$ denotes the element-wise matrix $l_1$ norm.

## 4 Experiments

We evaluate ALIGNED on multiple large-scale perturbation datasets for predicting genome-wide responses and assess the knowledge refinement mechanism in isolation[1]. The experiments address the following research questions:

Q1 Can ALIGNED achieve a higher balanced consistency than existing methods without damaging either data or knowledge consistency?

Q2 Is the knowledge refinement mechanism capable of re-discovering biologically meaningful and well-structured regulatory knowledge?

Q3 Does the framework leverage knowledge to improve prediction on unseen data, particularly under limited data availability?

## 4.1 Perturbation Prediction on Benchmark Datasets (Q1)

We focused on multiple large-scale perturbation datasets that are widely adopted for this prediction task: (i) Norman et al. (2019) for human K562 cells, including gene expression profiles under single and double perturbations across 102 genes (128 double and 102 single perturbations) with 89,357 samples; (ii) Dixit et al. (2016) for mouse BDMC cells, containing 19 single gene perturbations with 43,401 samples; and (iii) Adamson et al. (2016) for human K562 cells, containing 82 single gene perturbations with 65,899 samples. ALIGNED

---

[1]Details of experiment hardware and runtime are in Section E.2.

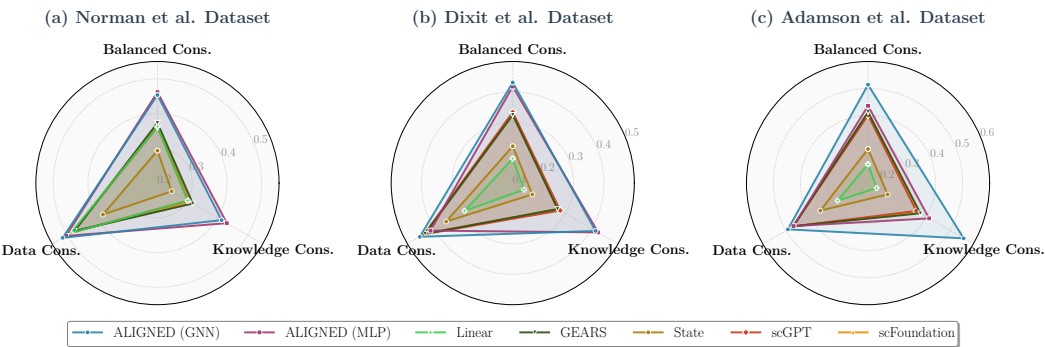

Figure 3: Balanced, data and knowledge consistency across methods.

predicts perturbation response profiles for all genes in the datasets. Our knowledge base $\mathcal{KB}$ integrates the Omnipath GRN (Türei et al., 2016) and the GO-based gene interaction graph from Roohani et al. (2024), covering 3,949 genes for the Norman et al. (2019) dataset and 2,958 genes for the Dixit et al. (2016) dataset. To evaluate methods on unseen perturbations, we split the test set of Norman et al. (2019) dataset to include 19 unseen single-gene perturbations and 18 unseen double-gene perturbations. The Dixit et al. (2016) and Adamson et al. (2016) datasets were split randomly.

We evaluated ALIGNED variants built with an MLP or a $\mathcal{KB}$-embedded GNN as the neural component. Performance of ALIGNED was compared in Figure 3 with state-of-the-art methods including: (i) GEARS, a GNN-based data-knowledge hybrid model (Roohani et al., 2024); (ii) foundation models scGPT (Cui et al., 2024) and scFoundation (Hao et al., 2024); (iii) a transformer-based model State (Adduri et al., 2025); (iv) a linear additive perturbation model incorporating regulatory knowledge (Ahlmann-Eltze et al., 2025). To ensure that all methods are measured on the same knowledge base, ALIGNED does not perform knowledge refinement during this comparison.

The ALIGNED framework with GNN and MLP results in Figure 3 performed adaptive neuro-symbolic alignment, but not knowledge refinement. For each method, we evaluated (i) data consistency $F_1(\bar{\boldsymbol{Y}}, \boldsymbol{Y}_{test})$ which measures predictive performance; (ii) knowledge consistency $F_1(\bar{\boldsymbol{Y}}, \delta_{\mathcal{KB}}(\boldsymbol{X}))$ which quantifies the extent to which the model remains grounded in curated biological prior; and (iii) the balanced consistency metric $F_{1\,balance}$ defined as Equation 3 evaluating the trade-off between inconsistent sources.

**Observation 1.** In Figure 3, ALIGNED achieved significantly higher knowledge consistency than other methods, with slightly higher data consistency. It consequently outperformed existing methods in balanced consistency. This shows ALIGNED's ability to make a better trade-off between inconsistent data and knowledge, enabling the framework to provide mechanistic understandings for black-box neural predictions.

**Observation 2.** In Figure 5, after one round of alignment and refinement, ALIGNED improved knowledge consistency significantly while keeping comparable data consistency. This further demonstrates that ALIGNED had learned an effective adaptor function to trade off data- and knowledge-derived information.

## 4.2 Knowledge Refinement of Gene Regulatory Networks (Q2)

In this section, we aim to answer whether ALIGNED can re-discover biologically meaningful and well-structured knowledge. We tested ALIGNED's knowledge refinement in isolation and evaluated the refined GRN interactions in three aspects: accuracy (Figure 4a), topology (Figure 4b) and pathway enrichment (Figure 4c).

We used the accuracy of interactions to test if ALIGNED's knowledge refinement can re-discover underlying regulations from synthetic data generated from OmniPath GRN (Türei et al., 2016). For topology, we evaluated the method's ability to produce well-structured

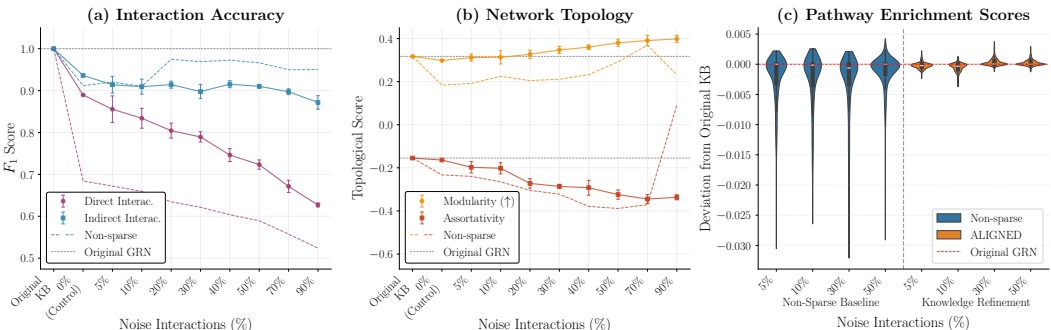

Figure 4: GRN knowledge refinement performance with ALIGNED.

GRNs, in terms of: (i) network modularity, for clustering quality of functional modules (Alon, 2007) and (ii) degree assortativity, for regulatory hub structures (Segal et al., 2003). In addition, we examined the method's ability to re-discover biologically meaningful interactions by cross-referencing external pathway databases. We compared overlaps with refined pathways using a gene set recovery algorithm (Huang et al., 2018) to obtain pathway enrichment scores.

The original OmniPath GRN (Türei et al., 2016) contains 2,958 genes and 113,056 regulatory interactions. We corrupted the original GRN by randomly adding and removing equal numbers of interactions at different noise levels, ranging from 5% to 90%, to simulate varying degrees of knowledge base errors. The experiment aimed to recover the original GRN from its synthetic data, using our knowledge refinement method initialized with the noisy GRN.

The non-sparse regularized baseline ablated the refinement mechanism in Equation 6, replacing the $l_1$ sparse regularization term with Frobenius regularization. Existing approaches, such as GRN inference, treat the knowledge base as static instead of performing incremental refinement, and therefore are not suitable for the comparison.

The accuracy was measured in $F_1$ score on both direct and indirect interactions (defined as Equation 1) of refined GRNs, assuming the original GRN as ground-truth. Network modularity and assortativity were measured on direct interactions of the GRN, with higher modularity scores for better clustering quality, and assortativity is usually negative in GRNs with well-structured regulatory hubs. To show the method's ability in re-discovering biologically meaningful interactions, we took 302 pathways from the KEGG pathway database (Kanehisa et al., 2025) as a cross-reference for gene set recovery, and measured the difference of recovery scores between reconstructed and original GRN for each pathway.

**Observation 1.** In Figure 4a, the accuracy of refined interactions by ALIGNED remained high ($F_1 > 0.7$) even with up to 40% noise. This shows that underlying regulatory knowledge in synthetic data can be captured by ALIGNED.

**Observation 2.** In Figure 4b, up to 20% noise, the topological measurements of the refined interactions are similar to those from the original GRN. This demonstrates the ability of ALIGNED in producing well-structured refined GRNs.

**Observation 3.** In Figure 4c, there are no significant differences of enrichment scores between the original and refined GRNs in most pathways. This indicates that ALIGNED can re-discover biologically meaningful knowledge annotated in cross-reference databases.

### 4.3 Perturbation Prediction on Bacterial Genome (Q3)

We evaluate our method on the *Escherichia coli* (*E. coli*) K-12 MG1655 strain using a combined dataset that includes 70 knockout perturbations by Lamoureux et al. (2023) (comprising 4 triple, 7 double, and 59 single perturbations, totaling 433 samples); and 7 data series with 16 single overexpression perturbations (73 samples) from the NCBI sequence read archive (Sayers et al., 2025). The knowledge base is constructed from the EcoCyc GRN

Table 1: Performance of ALIGNED on *E.coli* genome. The symbol "A" indicates an alignment stage, and "R" represents a refinement stage. ALIGNED with "A-R" executes alignment and then refinement, whereas "A-R-A" performs an additional alignment step.

| Model | ALIGNED Stages | Data Cons. | Knowledge Cons. | Balanced Cons. |
|---|---|---|---|---|
| MLP | MLP only | $0.3312_{\pm0.0004}$ | $0.3371_{\pm0.0009}$ | $0.3341_{\pm0.0006}$ |
| GNN | GNN only | $0.3773_{\pm0.0026}$ | $0.3605_{\pm0.0024}$ | $0.3689_{\pm0.0012}$ |
| ALIGNED (MLP) | A | $0.3872_{\pm0.0009}$ | $0.3708_{\pm0.0006}$ | $0.3784_{\pm0.0002}$ |
| | A-R | $0.3872_{\pm0.0009}$ | $0.3836_{\pm0.0067}$ | $0.3851_{\pm0.0033}$ |
| | A-R-A | $0.3812_{\pm0.0029}$ | $0.4025_{\pm0.0020}$ | $0.39404_{\pm0.0018}$ |
| | A-R-A-R | $0.3812_{\pm0.0029}$ | $0.4045_{\pm0.0038}$ | $0.3912_{\pm0.0013}$ |
| ALIGNED (GNN) | A | $0.3876_{\pm0.0060}$ | $0.3714_{\pm0.0269}$ | $0.3800_{\pm0.0159}$ |
| | A-R | $0.3876_{\pm0.0060}$ | $0.4520_{\pm0.0070}$ | $0.4130_{\pm0.0059}$ |
| | A-R-A | $0.3878_{\pm0.0064}$ | $0.4668_{\pm0.0235}$ | $0.4124_{\pm0.0042}$ |
| | A-R-A-R | $\mathbf{0.3878}_{\pm0.0064}$ | $\mathbf{0.5348}_{\pm0.0069}$ | $\mathbf{0.4288}_{\pm0.0063}$ |

(Moore et al., 2024), covering 315 regulator genes and 3,004 regulated genes. To evaluate generalization on unseen instances, we split the test set of unseen perturbations including 4 single overexpressions, 5 double knockouts, and 2 triple knockouts.

We tracked performance through a progressively richer procedure of ALIGNED iterations, including the neural-only models trained solely on labelled data and two complete iterations of neuro-symbolic alignment and knowledge refinement.

**Observation 1.** In Table 1, performance of prediction, i.e. data consistency, was significantly improved on unseen perturbations, simultaneously improving knowledge and balanced consistency. This indicates ALIGNED's ability of effectively leveraging knowledge-derived information under limited data availability.

## 4.4 Ablations

In Figure 5, we compare three progressively richer variants: (i) neural-only model with no symbolic refinement (GNN only) (ii) adaptive neuro-symbolic alignment without refinement ("A") and (iii) the complete ALIGNED framework ("A-R"). Setting (i) is a GNN trained solely on labelled data, with no symbolic alignment or refinement. Setting (ii) adds the alignment procedure on top of the neural-only baseline, but without updating the knowledge base. Setting (iii) includes both adaptive alignment and knowledge refinement steps. These ablations isolate the contribution of each component. We observe that adaptive alignment alone improves knowledge consistency, showing that the adaptor effectively incorporates information from the symbolic component. In addition, symbolic refinement provides a substantial additional improvement in knowledge consistency, while maintaining accuracy comparable to the neural-only model.

Table 2 reports ablations on $\mathcal{KB}$ and the adaptor loss (Equation 4) using the Norman et al. dataset and a GNN as the neural component. We compare ALIGNED with: (i) a randomly structured GRN as $\mathcal{KB}$; (ii) removal of the inconsistency term $\text{Inc}(\boldsymbol{a}, \boldsymbol{x}, \hat{\boldsymbol{y}}, \mathcal{KB})$; (iii) removal of the knowledge-usage restriction term $L_{len}(\boldsymbol{a})$; and (iv) a random weight vector $\boldsymbol{w}$. The results show: (i) without a meaningful prior knowledge, the adaptor defaults to neural predictions; (ii) the inconsistency term encourages stronger alignment with knowledge, at a small cost in predictive accuracy; (iii) the usage-restriction term prevents over-reliance on symbolic predictions; and (iv) the weight vector helps balance predictive performance with appropriate knowledge usage.

## 5 Related Work

**Perturbation Response Prediction.** Recent approaches fall into two categories: methods that utilize the compositional nature of genetic perturbations in learning latent representations (Lotfollahi et al., 2023; Cui et al., 2024; Hao et al., 2024), hybrid methods that

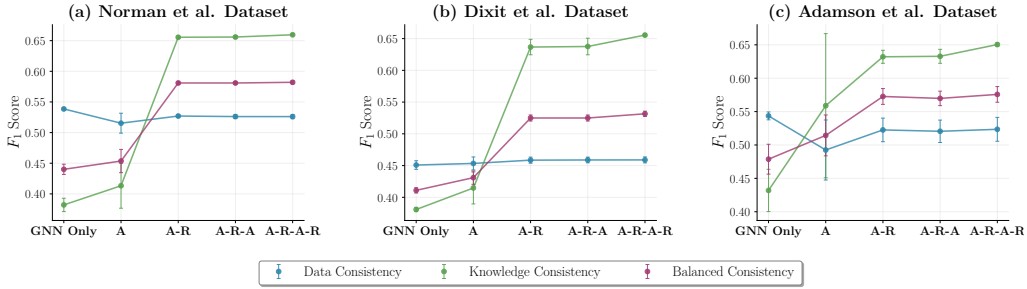

Figure 5: Performance of the complete ALIGNED framework built with GNN. The symbol "A" indicates an alignment stage, and "R" represents a refinement stage. ALIGNED with "A-R" executes alignment and then refinement, whereas "A-R-A" performs an additional alignment step.

Table 2: Ablation of $\mathcal{KB}$ and $L_a$. "A" indicates ALIGNED performs one alignment step.

| Ablation Settings | ALIGNED Stages | Data Cons. | Knowledge Cons. | Balanced Cons. |
|---|---|---|---|---|
| Default | A | 0.5101 | 0.4172 | 0.4567 |
| Random $\mathcal{KB}$ | A | 0.5611 | 0.3190 | 0.3921 |
| Remove Inc | A | 0.5399 | 0.3469 | 0.4127 |
| Remove $L_{len}$ | A | 0.5047 | 0.4442 | 0.4715 |
| Random $w$ | A | 0.4074 | 0.6194 | 0.4813 |

leverage prior knowledge from biological networks (Roohani et al., 2024; Wenkel et al., 2025; Littman et al., 2025) or textual embeddings (Wang et al., 2024). In contrast, ALIGNED does not assume GRNs to be static and can systematically refine GRNs by adaptive learning from datasets and knowledge bases to leverage reliable information.

**Neuro-Symbolic Learning.** The Abductive Learning (ABL) framework (Zhou, 2019) integrates deep learning with symbolic constraints through consistency optimization. Extensions include $Meta_{abd}$ (Dai & Muggleton, 2021) for visual-symbolic reasoning, $ABL_{NC}$ for knowledge refinement (Huang et al., 2023), and $ABL_{refl}$ (Hu et al., 2025) for efficient neuro-symbolic integration using reinforcement learning mechanisms. Additionally, Cornelio et al. (2023) proposed a learnable trade-off mechanism between data and knowledge sources. In comparison, ALIGNED does not assume the knowledge base to be fixed or absolutely correct. Since both biological data and curated knowledge exhibit domain-specific noise and incompleteness, ALIGNED jointly aligns neural and symbolic predictions to leverage the most reliable signals and actively refines knowledge bases during training.

## 6 Conclusion and Future Work

In this work, we introduced ALIGNED, a novel end-to-end framework that achieves balanced neuro-symbolic alignment and knowledge refinement for predicting genetic perturbation. Importantly, our work not only enhances transparency about the biological relationships behind predictions but also enables the evolution of biological knowledge from large-scale datasets, advancing beyond current black-box approaches.

While we acknowledge the limitations in our regulatory network modelling, alternative methods (Covert et al., 2004; Stoll et al., 2017; Abou-Jaoudé et al., 2016) face significant scalability issues. Future work could explore differentiable models (Faure et al., 2023) and refine them with experimental data. Furthermore, the ALIGNED framework only assumes the availability of a neural predictor and a symbolic component with consistency measurement. Therefore, ALIGNED can be extended to different biological tasks by leveraging other prior knowledge, such as protein-protein interaction networks (Rodriguez-Mier et al., 2025) and metabolic networks (Faure et al., 2023).

ACKNOWLEDGMENTS

We thank our colleagues at the European Bioinformatics Institute (EMBL-EBI), Julio Saez-Rodriguez, Aurelien Dugourd, Ricardo O. Ramirez-Flores, Pablo Rodríguez Mier, Philipp Schaefer, and Daniele Bottazzi, for their valuable feedback and discussions throughout the development of this work; our colleagues at Nanjing University, Yu-Feng Li, Xin-Hao Zhu, Jun-Tong Wang, and Jing He, for their discussions and suggestions throughout the initialization and implementation of this work; our collaborators at Imperial College London, Geoff S. Baldwin, Shi-Shun Liang, and Alfie J. Brown on discussions and directions on biological applications.

REPRODUCIBILITY

Details of experiments and reproducibility are included in Section E. The code repository is available at https://github.com/yfxiang0112/Aligned.

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

## A  Usage of Large Language Models

We used Claude and ChatGPT mainly to polish the language after all intellectual content has been drafted, along with Grammarly as a language editing tool.

## B  Details of the Knowledge Base

### B.1  Formal Definition

With regard to positivity of edges, the GRN can be represented as a datalog program (Minker, 1988) with two predicates, $regulates_+/2$ and $regulates_-/2$, where genes are constant names. We use the transitive closure of $regulates_+$ and $regulates_-$ as the knowledge base

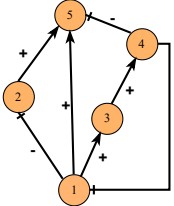

Figure 6: A 5-node GRN example.

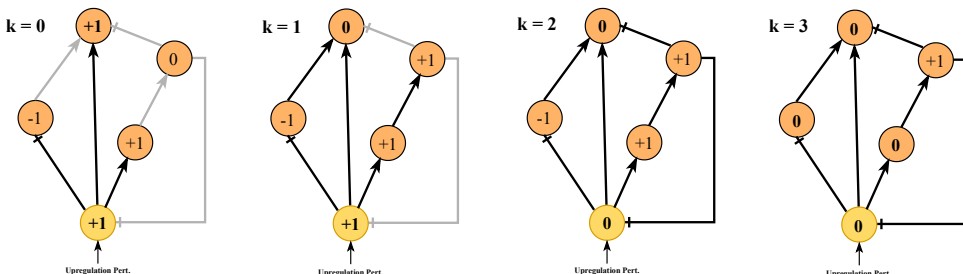

Figure 7: A demonstration of approximative fixpoint.

in the reasoning component of our framework, denoted as $r_+/2$ and $r_-/2$. The transitive closure can be evaluated as a bilinear recursive program (Ioannidis & Wong, 1991):

$$
\begin{aligned}
r_+ &\leftarrow (regulates_+(a,c) \wedge r_+(c,b)) \vee (regulates_-(a,c) \wedge r_-(c,b)) \\
r_- &\leftarrow (regulates_+(a,c) \wedge r_-(c,b)) \vee (regulates_-(a,c) \wedge r_+(c,b)) \\
r_+(a,b) &\leftarrow regulates_+(a,b) \\
r_-(a,b) &\leftarrow regulates_-(a,b)
\end{aligned} \tag{7}
$$

The transitive closure represents all regulatory pathways, accounting for the indirect effects of positive and negative regulation. The recursive program states that all direct positive (negative) regulations are included in the positive (negative) transitive closure, while an indirect pathway containing an even number of negative regulations contributes to the positive closure, and an odd number of negative regulations contributes to the negative closure. The datalog program can then be compiled as a recursive Boolean matrix multiplication (Equation 1), where matrices of positive (negative) direct regulations $\boldsymbol{R}_+^{(0)}, \boldsymbol{R}_-^{(0)} \in \{0,1\}^{n \times n}$ are compiled from $regulates_+, regulates_-$:

$$
(R_+^{(0)})_{ij} = \begin{cases} 1, & regulates_+(i,j) \\ 0, & \text{otherwise} \end{cases} \quad , \quad (R_-^{(0)})_{ij} = \begin{cases} 1, & regulates_-(i,j) \\ 0, & \text{otherwise} \end{cases}
$$

and indirect regulation matrices $\boldsymbol{R}_+^{(k)}, \boldsymbol{R}_-^{(k)}$ are computed from Equation 1.

### B.2 DEMONSTRATION

For a 5 nodes example GRN in Figure 6, a simple demonstration of the approximative fixpoint of regulatory interactions (Equation 1) is shown as Figure 7.

In this example, $\boldsymbol{R}_+^{(0)}, \boldsymbol{R}_-^{(0)}$ are compiled as:

$$
\boldsymbol{R}_+^{(0)} = \begin{bmatrix} 1 & 0 & 1 & 0 & 1 \\ 0 & 1 & 0 & 0 & 1 \\ 0 & 0 & 1 & 1 & 0 \\ 0 & 0 & 0 & 1 & 0 \\ 0 & 0 & 0 & 0 & 1 \end{bmatrix}, \boldsymbol{R}_-^{(0)} = \begin{bmatrix} 0 & 1 & 0 & 0 & 0 \\ 0 & 0 & 0 & 0 & 0 \\ 0 & 0 & 0 & 0 & 0 \\ 1 & 0 & 0 & 0 & 1 \\ 0 & 0 & 0 & 0 & 0 \end{bmatrix},
$$

and the symbolic prediction when $k = 2$ is:

$$\delta_{\langle \boldsymbol{R}_+^{(2)}, \boldsymbol{R}_-^{(2)} \rangle}(\boldsymbol{x}) = \left( \begin{bmatrix} 1 & 0 & 1 & 1 & 1 \\ 0 & 1 & 0 & 0 & 1 \\ 0 & 1 & 1 & 1 & 0 \\ 0 & 1 & 0 & 1 & 1 \\ 0 & 0 & 0 & 0 & 1 \end{bmatrix} - \begin{bmatrix} 1 & 1 & 0 & 0 & 1 \\ 0 & 0 & 0 & 0 & 0 \\ 1 & 0 & 1 & 0 & 1 \\ 1 & 0 & 1 & 1 & 1 \\ 0 & 0 & 0 & 0 & 0 \end{bmatrix} \right)^{\top} \begin{bmatrix} 1 \\ 0 \\ 0 \\ 0 \\ 0 \end{bmatrix} = \begin{bmatrix} 0 \\ -1 \\ 1 \\ 1 \\ 0 \end{bmatrix}$$

And $\delta_{\langle \boldsymbol{R}_+^{(k)}, \boldsymbol{R}_-^{(k)} \rangle}(\boldsymbol{x}) = \boldsymbol{0}$ for $k \leq 4$ in this example, due to the negative feedback loop.

### B.3 Discussion

Our modelling of regulatory effect given by Equation 7 does not assume that activating and inhibitory effects combine through simple additive rules. The formulation is path-dependent and encodes how positive and negative regulations compose across multi-step pathways. However, such modelling is still not enough to describe the biological reality, and future work could further explore the other differentiable modelling approaches of genome-scale GRN under the ALIGNED framework.

In actual experiments in Section 4.1, we introduced an assumption that up/down regulation of a node can be decided by its in-degree of positive and negative interactions, in order to capture more detailed regulatory behaviours. This is achieved by using an integer variant of Equation 1. In this setting, the value of node 5 in Figure 7 will be -1 when $k = 2$, i.e. $\delta_{\langle \boldsymbol{R}_+^{(2)}, \boldsymbol{R}_-^{(2)} \rangle}(\boldsymbol{x}) = [0, -1, 1, 1, -1]^{\top}$, and the network behaviour will be more complicated in complex networks.

## C Discussions of Convergence

The convergence of the ALIGNED iterations is guaranteed under certain theoretical assumptions. The inconsistency measurement defined in Equation 2 is non-negative, lower-bounded by 0. Since both the neural and the symbolic component are updated with the neuro-symbolic prediction $\langle \boldsymbol{X}_u, \bar{\boldsymbol{Y}}_u \rangle$ in each iteration, the inconsistency measurement reduces if the following assumptions hold: (a) the neural prediction $\hat{\boldsymbol{Y}}_u$ aligns better with $\bar{\boldsymbol{Y}}_u$ after update, and (b) the symbolic prediction $\delta_{\mathcal{KB}}(\boldsymbol{X}_u)$ aligns better with $\bar{\boldsymbol{Y}}_u$ after the knowledge refinement. Therefore, the inconsistency decreases monotonically until it reaches a fixed point, where further updates to either module no longer reduce disagreement and the ALIGNED reaches convergence.

In practice, ALIGNED also converges when the assumptions (a), (b) do not strictly hold in theory. Empirically, the iterations are observed to stabilize within a small number of iterations (typically 2-3), as shown in Figure 5.

## D Framework Overview

Additional demonstrations for the ALIGNED framework, including an overview figure Figure 8 and pseudo-code Algorithm 1, are included here.

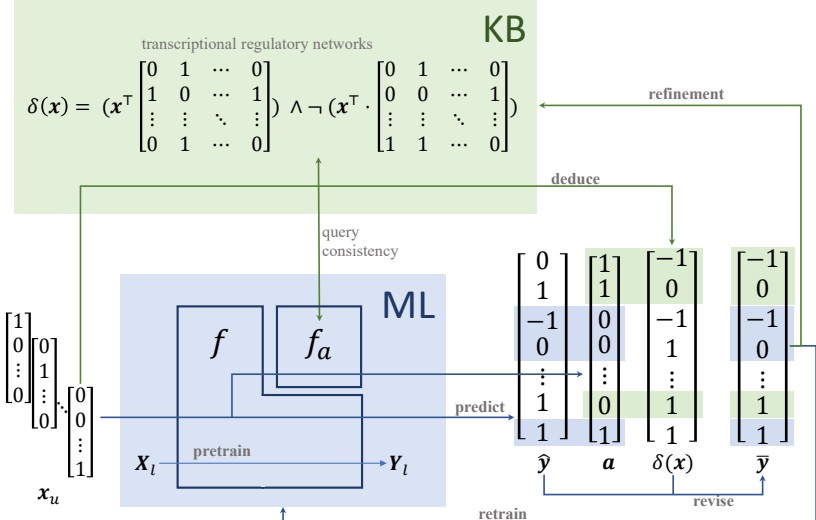

Figure 8: The ALIGNED framework. The nueral component is marked in blue, and the symbolic component ingreen.

---

**Algorithm 1:** ALIGNED Framework

**input** : Labelled dataset $D_l = (\boldsymbol{X}_l, \boldsymbol{Y}_l)$; Unlabelled dataset $D_u = \boldsymbol{X}_u$;
Gene interactions$\langle \boldsymbol{R}_+^{(0)}, \boldsymbol{R}_-^{(0)} \rangle$; Iteration limit $T$
**output:** Trained neural model $(f_y, f_a)$;
Updated knowledge base $\mathcal{KB}$

1   $\mathcal{KB} \leftarrow \langle \boldsymbol{R}_+^{(k)}, \boldsymbol{R}_-^{(k)} \rangle$;
2   $(f_y, f_a) \leftarrow \text{train}(\boldsymbol{X}_l, \boldsymbol{Y}_l, \mathcal{KB})$;
3   **for** $1 \le t \le T$ **do**
4     $\hat{\boldsymbol{Y}} \leftarrow f_y(\boldsymbol{X}_u)$;
5     $\boldsymbol{A} \leftarrow f_a(\boldsymbol{X}_u)$;
6     $\delta_{\mathcal{KB}} \leftarrow \boldsymbol{X}_u(\boldsymbol{R}_+^{(k)} - \boldsymbol{R}_-^{(k)})$;
7     $\bar{\boldsymbol{Y}} \leftarrow \begin{cases} \bar{Y}_{ij} = \hat{Y}_{ij}, & A_{ij} = 0 \\ \bar{Y}_{ij} = \delta_{\mathcal{KB}}(\boldsymbol{X}_u)_{ij}, & A_{ij} = 1 \end{cases}$;
8     $(f_y, f_a) \leftarrow \text{update}(\boldsymbol{X}_u, \bar{\boldsymbol{Y}}_u)$;
9     $\mathcal{KB} \leftarrow \text{refine}(\boldsymbol{X}_u, \bar{\boldsymbol{Y}}_u, \mathcal{KB})$
10 **end**

---

## E   EXPERIMENT DETAILS AND REPRODUCIBILITY

### E.1   REPRODUCIBILITY AND CODE AVAILABILITY

Unless specified, our experiments on ALIGNED and other methods used random seeds to split for training, validation and test set.

### E.2   HARDWARE AND RUNTIME

All experiments were conducted using a single Intel Xeon Gold 6342 CPU (2.80 GHz, 32 GB memory) and a single NVIDIA A100 GPU (80 GB). Training ALIGNED took an average of 10-12 hours per run. Under our experimental setup, the training time of ALIGNED is manageable given the size of the GRN and the complexity of the refinement procedure.

### E.3 Hyperparameters

In Section 4.1, key hyperparameters of our ALIGNED method as follows:

$k = 4$ in Equation 1.

$$\boldsymbol{w} = 0.5\Big(\sum_{\boldsymbol{x},\boldsymbol{y}\in\langle\boldsymbol{X},\boldsymbol{Y}\rangle}\mathbf{1}_{\boldsymbol{y}=\delta_{\mathcal{KB}}(\boldsymbol{x})}/|\boldsymbol{X}|\Big) + 0.2\Big(\sum_{1\leq i\leq n}(R^{(0)}_{+,i,j}+R^{(0)}_{-,i,j})/2n\Big) + 0.3\boldsymbol{c}_{GO} \text{ in Equation 4,}$$

tuned on Norman et al. (2019) dataset and Türei et al. (2016) knowledge base, where $\langle\boldsymbol{X},\boldsymbol{Y}\rangle$ is the training dataset, $\mathcal{KB} = \langle\boldsymbol{R}^{(k)}_{+},\boldsymbol{R}^{(k)}_{-}\rangle$ is the knowledge base, $n$ is the total number of genes. $\boldsymbol{c}_{GO}$ is the number of GO annotations for each gene, normalized to $[0,1]$.

The weight vector $\boldsymbol{w}$ is computed separately for each dataset, but always using the same fixed procedure above. In particular, the component of $\boldsymbol{w}$ that reflects data-knowledge inconsistency is derived only from the training split, while the other components encode priors from the structured knowledge bases. This makes $\boldsymbol{w}$ a dataset-specific but non-hand-tuned quantity.

$\gamma = 5$ in Equation 3; $C_w = 10$, $C_l = 5$, $\theta = 0.3|\boldsymbol{a}|$ in Equation 4; $C = 10$ in Equation 5; $\lambda = 1$, $t_0 = 100$ and $t_k = 1$ in Equation 6.

### E.4 Experiments on Hyperparameters of the Adaptor

To study the effect on ALIGNED's overall performance of each hyperparameter, we conducted a comparison of different parameter selection of $k$ in Equation 1, $\theta$ in Equation 4, $t_0, t_k$ in Equation 6, as well as different computing for $\boldsymbol{w}$ in Equation 4. The comparison in Table 3 was ran for 2 ALIGNED loops, and we report the performance at neuro-symbolic alignment of the first loop (Align 1) since the hyperparameters put most effect on the behavior of the adaptor, as well as the performance at the end of the loops (Refine 2).

All comparisons were performed on the Norman et al. (2019) dataset with GNN as the neural component, including the default setting in Section 4.1 (denoted as "default", with $k = 4$, $\theta = 0.3$ and $\boldsymbol{w}$ computed as Section E.3), different values of $k, \theta$, and different coefficients in computing $\boldsymbol{w}$ ("term $i$ +" means that the $i$-th term's coefficient in Section E.3 $\boldsymbol{w}$'s formula is increased by 0.1, following by clipping the $\boldsymbol{w}$'s elements to keep in $[0,1]$).

The results in Table 3 indicate that, maximum allowed pathway length ($k$) in the modelling of $\mathcal{KB}$ is positively correlated with data consistency (predictive performance on test data), but negatively correlated with the knowledge consistency. The threshold of $\mathcal{KB}$ usage ($\theta$) has a minor effect on encouraging $\mathcal{KB}$ usage with higher $\theta$. Term coefficients in $\boldsymbol{w}$'s formula has a positive effect on balanced consistency, in comparison with setting all elements of $\boldsymbol{w}$ to 0.5. Every term in $\boldsymbol{w}$'s formula is positively correlated with knowledge consistency but slightly compromises data consistency.

### E.5 Experiments on Hyperparameters of Knowledge Refinement

We conducted experiment in Section 4.2 to examine the effect of the approximation coefficient $t_k$ and $t_0$ in Equation 6. The default setting in Section 4.2 ($t_0 = 100, t_k = 1$) is denoted as "Default", and the performance was compared in $F_1$ score of direct interactions, same with Figure 4a. As shown in Table 4, the performance of knowledge refinement is positively correlated with $t_0$ and $t_k$. Larger values of $t_0$ and $t_k$ than default have diminishing improvement returns in performance. Generally, larger values of $t_0$ and $t_k$ require higher computational resource. Therefore, we decided on the current default values based on both performance and computational cost.

### E.6 Dataset Split

The detail of train-test data split in Experiment of Section 4.1 is shown in Table 5.

Table 3: Performance of ALIGNED on different hyperparameter selections. The symbol "A" indicates an alignment stage, and "R" represents a refinement stage. ALIGNED with "A" executes one alignment step, whereas "A-R-A-R" performs two rounds of alignment and refinement steps.

| ALIGNED Stages | Operation | Data Cons. | Knowledge Cons. | Balanced Cons. |
|---|---|---|---|---|
| GNN only | default | 0.5344 | 0.3823 | 0.4397 |
| A | default | 0.5101 | 0.4172 | 0.4567 |
| | $k = 0$ | 0.4426 | 0.3270 | 0.3719 |
| | $k = 2$ | 0.4960 | 0.4545 | 0.4738 |
| | $k = 6$ | 0.5129 | 0.4159 | 0.4568 |
| | $\theta = 0.1$ | 0.5148 | 0.4051 | 0.4502 |
| | $\theta = 0.5$ | 0.5047 | 0.4442 | 0.4715 |
| | $\boldsymbol{w = 0.5}$ | 0.3944 | 0.6381 | 0.4744 |
| | $\boldsymbol{w}$: term 1 + | 0.4993 | 0.6577 | 0.5624 |
| | $\boldsymbol{w}$: term 2 + | 0.5120 | 0.4335 | 0.4678 |
| | $\boldsymbol{w}$: term 3 + | 0.4999 | 0.4294 | 0.4606 |
| A-R-A-R | default | 0.5233 | 0.6570 | 0.5788 |
| | $k = 0$ | 0.4377 | 0.3388 | 0.3788 |
| | $k = 2$ | 0.5115 | 0.6416 | 0.5656 |
| | $k = 6$ | 0.5028 | 0.6606 | 0.5658 |
| | $\theta = 0.1$ | 0.5254 | 0.6549 | 0.5795 |
| | $\theta = 0.5$ | 0.5175 | 0.6607 | 0.5761 |
| | $\boldsymbol{w = 0.5}$ | 0.4047 | 0.6476 | 0.4853 |
| | $\boldsymbol{w}$: term 1 + | 0.5003 | 0.6509 | 0.5609 |
| | $\boldsymbol{w}$: term 2 + | 0.5226 | 0.6614 | 0.5798 |
| | $\boldsymbol{w}$: term 3 + | 0.5099 | 0.6582 | 0.5700 |

Table 4: Performance of knowledge refinement on different hyperparameter selections

| Noisy Interactions | Default | $t_0 = 1000, t_k = 10$ | $t_0 = 10, t_k = 0.1$ |
|---|---|---|---|
| 0% | 0.8896 | 0.9052 | 0.6726 |
| 5% | 0.8554 | 0.8895 | 0.6262 |
| 10% | 0.8339 | 0.8753 | 0.5921 |
| 30% | 0.7894 | 0.8181 | 0.5347 |
| 50% | 0.7233 | 0.7604 | 0.5357 |

# F   ADDITIONAL RESULTS

## F.1   CURATION EFFORT AND $\mathcal{KB}$ PREDICTION CONTRIBUTION

We conducted an experiment to stratify edges in OmniPath by their curation effort (a proxy for knowledge reliability) and measured how often the adaptor selects the symbolic versus neural predictions in each regime. ALIGNED did not have access to this curation effort information during training. Our results in Table 6 show that the adaptor automatically down-weights the symbolic predictions when the GRN is less reliable (low curation effort) and relies more heavily on the neural predictor in those cases. Conversely, when the edges in $\mathcal{KB}$ are better curated, the adaptor increases its symbolic usage.

Table 5: Test dataset split in Experiment of Section 4.1

| Dataset | Replicates | Test set perturbations |
|---------|------------|------------------------|
| Norman | - | FEV, TBX3, AHR, S1PR2, TBX2, MAP7D1, ZBTB1, OSR2, ISL2, BPGM, SGK1, MAP7D1, PRDM1, UBASH3B + OSR2, PRDM1 + CBFA2T3, SGK1 + TBX3, TBX3 + TBX2, PTPN12 + OSR2, BPGM + SAMD1, AHR + KLF1, FEV + MAP7D1, SAMD1 + ZBTB1, CEBPB + OSR2, SGK1 + S1PR2, FEV + ISL2, SGK1 + TBX2, ETS2 + MAP7D1, AHR + FEV, FEV + CBFA2T3, FOSB + OSR2, BPGM + ZBTB1 |
| Dixit | 1 | ELF1, E2F4, CIT, CEP55, AURKA |
| | 2 | ELK1, NR2C2, ECT2, CENPE |
| | 3 | ELK1, E2F4 |
| | 4 | CENPE, TOR1AIP1, RACGAP1 |
| | 5 | GABPA, E2F4, CENPE |
| Adamson | 1 | DDIT3, OST4, SEC61A1, HARS, IARS2, SOCS1, MRGBP, MTHFD1, TMEM167A, TTI2, SLC39A7, MRPL39, SEC63, MARS, ARHGAP22, EIF2S1, TIMM44, PTDSS1, CHERP, SRPRB, DHDDS, COPZ1, CARS, P4HB |
| | 2 | BHLHE40, DDIT3, SAMM50, MTHFD1, EIF2B2, PSMD4, EIF2S1, CCND3, PPWD1, COPZ1, EIF2B3, XRN1, TMED2, SPCS2 |
| | 3 | DASAMM50, MRGBP, TMEM167A, NEDD8, HSPA9, MRPL39, FARSB, HSD17B12, SEC61B, ASCC3, ARHGAP22, HSPA5, CHERP |
| | 4 | ZNF326, DDIT3, OST4, QARS, MRGBP, MRPL39, SEC63, MANF, HSD17B12, SARS, TIMM44, DDOST, SLC35B1, CCND3, EIF2B3, XRN1, TMED2, SPCS3 |
| | 5 | ZNF326, BHLHE40, SEC61A1, IER3IP1, YIPF5, IARS2, ATP5B, MTHFD1, GNPNAT1, TARS, PSMD4, CHERP, DHDDS, DNAJC19 |

Table 6: Knowledge vs. data usage across curation effort levels

| Curation Effort | Knowledge Usage | Data Usage |
|-----------------|-----------------|------------|
| 0 | 13.3% | 86.7% |
| [1,5) | 24.7% | 75.3% |
| [5,10) | **31.9%** | 68.1% |
| $\geq 10$ | 31.7% | 68.3% |

## F.2 GRN FAILURE MODES

We experimented using multiple external knowledge sources built by Chevalley et al. (2025), include STRINGdb (PPI), CORUM (PPI) and a ChipSeq network derived from Chip-Atlas and ENCODE, as independent cross-references.

We stratified gene pairs into three categories: (i) not regulated in both (sparse), (ii) regulated in both (high confidence), (iii) regulated only in our $\mathcal{KB}$ or in reference (low confidence). The rationale is that interactions supported by both our $\mathcal{KB}$ (OmniPath) and an external reference are more likely to be reliable, whereas interactions found in only one source may imply biases, incompleteness, or systematic errors. The results in Table 7 demonstrate that when coverage is sparse or inconsistent, ALIGNED relies heavily on the neural component

( 90%). This prevents the model from propagating uncertain symbolic signals. In addition, when prior knowledge is less reliable, the symbolic component contributes less to the overall predictions compared to high-confidence edges.

Table 7: Knowledge vs. data usage comparing with external databases

| Database | Edges | Knowledge Usage | Data Usage |
|---|---|---|---|
| STRINGdb | sparse | 10.5% | 89.5% |
| | high confidence | 74.6% | 25.4% |
| | low confidence | 54.8% | 45.2% |
| CORUM | sparse | 89.4% | 10.6% |
| | high confidence | 16.9% | 83.1% |
| | low confidence | 28.2% | 71.8% |
| ChipSeq | sparse | 89.4% | 10.6% |
| | high confidence | 25.4% | 74.6% |
| | low confidence | 29.4% | 70.6% |

### F.3 Interaction Recovery

We randomly sampled interactions (3 replicates) from our $\mathcal{KB}$ that simultaneously exists in a PPI database (STRING or Corum) and the ChipSeq network by Chevalley et al. (2025), which we believe that they tend to be true interactions with a high confidence. We removed them selected interactions from our $\mathcal{KB}$, ran ALIGNED using the $\mathcal{KB}$ with removed edges and examined whether the removed interactions can be recovered after 2 alignment / refinement iterations.

The selected interactions and the performance of recovery are listed in Table 8. The results show that the ALIGNED iterations are able to correct potential noise and incompleteness in $\mathcal{KB}$ using neural predictions.

Table 8: Recovery rate of confident interaction removal

| Replicate | Recovered/Total | Interactions |
|---|---|---|
| 1 | 11/12 | JUN - CTNNB1, SPI1 - FCGRT, CEBPA - SPI1, CEBPB - TGFB1, CEBPB - GDF15, CEBPB - TRIB3, CEBPB - HSP90AA1, CEBPB - RUN X2, CEBPA - CEBPE, JUN - BEX2, JUN - GSTP1, JUN - KRT18 |
| 2 | 10/10 | JUN - CTNNB1, JUN - GSTP1, CEBPB - TGFB1, CEBPB - CEBPA, CEBPB - TRIB3, CEBPB - HSP90AA1, CEBPB - DDIT3, CEBPB - GST P1, CEBPB - CSF3R, CEBPA - CEBPE |
| 3 | 10/11 | JUN - STMN1, JUN - MYC, JUN - GSTP1, JUN - VIM, JUN - KRT18, SPI1 - FCGRT, CEBPB - CSF3R, CEBPB - RUNX2, CEBPB - TRIB3, CEBPB - GDF15, CEBPA - CSF3R |

### F.4 Cross-Reference for Section 4.2

We extended the GRN refinement experiment in Section 4.2 with additional cross-reference comparisons using external databases built by Chevalley et al. (2025). Performance of the knowledge refinement is measured via comparing the accuracy of GRN after refinement contrasting with the reference database, in terms of $F_1$ score. The performance is compared with the non-sparse baseline (using Frobenius regularization), as noted in Section 4.2. Our results are shown in Table 9, The results demonstrate that our method is able to learn interactions that are verified by external reference sources.

Table 9: GRN after refinement contrasting with reference databases in terms of $F_1$

| Method | Noisy Interactions | STRINGdb | CORUM | ChipSeq |
|---|---|---|---|---|
| ALIGNED | 0% | 0.4561 | 0.0310 | 0.2015 |
| | 5% | 0.4406 | 0.0293 | 0.1814 |
| | 10% | 0.4271 | 0.0270 | 0.1764 |
| | 30% | 0.4093 | 0.0258 | 0.1714 |
| | 50% | 0.3706 | 0.0207 | 0.1363 |
| Non-sparse | 0% | 0.2644 | 0.0058 | 0.0372 |
| | 5% | 0.2600 | 0.0054 | 0.0350 |
| | 10% | 0.2685 | 0.0074 | 0.0530 |
| | 30% | 0.2351 | 0.0041 | 0.0315 |
| | 50% | 0.0472 | 0.0002 | 0.0018 |

### F.5 Shortcut Regulations in Section 4.2

To examine whether the knowledge refinement learns undesirable shortcut signals from the data, we extended the GRN-refinement experiment in Section 4.2 by comparing the proportion of direct versus indirect interactions after knowledge refinement on the noisy GRN. We evaluated how many direct and indirect interactions in the original GRN were learned, where direct interactions are desired and indirect interactions represents shortcut regulatory effects in the data.

The results in Table 10 show that ALIGNED's refinement predominantly recovers direct interactions, while indirect ones remain limited. In contrast, the non-sparse baseline (using Frobenius regularization) overfits to indirect interactions. These findings suggest that our sparse refinement strategy effectively guards against overfitting to shortcut signals.

Table 10: Direct vs. indirect interactions in Section 4.2

| Noisy Interactions | ALIGNED | | Non-sparse | |
|---|---|---|---|---|
| | Direct | Indirect | Direct | Indirect |
| 0% | 81.4% | 17.0% | 16.6% | 68.0% |
| 5% | 73.3% | 21.3% | 15.0% | 68.6% |
| 10% | 71.8% | 23.2% | 24.1% | 53.6% |
| 30% | 71.2% | 18.8% | 11.9% | 70.5% |
| 50% | 58.6% | 24.7% | 0.6% | 36.8% |

### F.6 Full Results of Consistency Benchmark on Norman et al. (2019); Dixit et al. (2016); Adamson et al. (2016) Datasets

Evaluations of Section 4.1 on train and test data in precision and recall are shown in Table 11. Full benchmark results of Section 4.1 are shown in Table 12.

Table 11: Precision and recall in Figure 3,5. "A" is an alignment stage, and "R" is a refinement stage. ALIGNED with "A-R" executes alignment and then refinement.

| Dataset | Model | ALIGNED Stage | Train Data | | Test Data | | Knowledge Base | |
|---|---|---|---|---|---|---|---|---|
| | | | Precision | Recall | Precision | Recall | Precision | Recall |
| Norman | MLP | MLP only | 0.556 | 0.544 | 0.543 | 0.541 | 0.568 | 0.427 |
| | | A | 0.510 | 0.539 | 0.503 | 0.537 | 0.593 | 0.465 |
| | | A-R | 0.553 | 0.535 | 0.537 | 0.534 | 0.644 | 0.652 |
| | GNN | GNN only | 0.552 | 0.556 | 0.552 | 0.553 | 0.576 | 0.433 |
| | | A | 0.524 | 0.544 | 0.523 | 0.543 | 0.597 | 0.458 |
| | | A-R | 0.553 | 0.538 | 0.545 | 0.540 | 0.653 | 0.655 |
| Dixit | MLP | MLP only | 0.498 | 0.449 | 0.494 | 0.448 | 0.708 | 0.476 |
| | | A | 0.474 | 0.445 | 0.454 | 0.445 | 0.599 | 0.483 |
| | | A-R | 0.503 | 0.446 | 0.462 | 0.441 | 0.590 | 0.645 |
| | GNN | GNN only | 0.493 | 0.489 | 0.514 | 0.505 | 0.604 | 0.502 |
| | | A | 0.480 | 0.499 | 0.480 | 0.504 | 0.563 | 0.525 |
| | | A-R | 0.485 | 0.498 | 0.484 | 0.499 | 0.616 | 0.638 |
| Adamson | MLP | MLP only | 0.559 | 0.463 | 0.524 | 0.467 | 0.587 | 0.448 |
| | | A | 0.537 | 0.464 | 0.486 | 0.467 | 0.533 | 0.460 |
| | | A-R | 0.571 | 0.461 | 0.496 | 0.457 | 0.562 | 0.643 |
| | GNN | GNN only | 0.561 | 0.555 | 0.578 | 0.579 | 0.577 | 0.502 |
| | | A | 0.511 | 0.528 | 0.531 | 0.566 | 0.676 | 0.634 |
| | | A-R | 0.557 | 0.543 | 0.553 | 0.560 | 0.633 | 0.645 |

Table 12: Consistency benchmark in Figure 3,5. "A" is an alignment stage, and "R" is a refinement stage. ALIGNED with "A-R" executes alignment and then refinement.

| Dataset | Model | ALIGNED Stages | Data Cons. | Knowledge Cons. | Balanced Cons. |
|---|---|---|---|---|---|
| Norman | linear | - | $0.4875_{\pm 0.0000}$ | $0.3009_{\pm 0.0000}$ | $0.3621_{\pm 0.0000}$ |
| | GEARS | - | $0.4755_{\pm 0.0063}$ | $0.3175_{\pm 0.0100}$ | $0.3731_{\pm 0.0066}$ |
| | State | - | $0.1440_{\pm 0.0001}$ | $0.3811_{\pm 0.0003}$ | $0.2473_{\pm 0.0002}$ |
| | Morph | - | $0.3378_{\pm 0.0000}$ | $0.2580_{\pm 0.0000}$ | $0.2900_{\pm 0.0000}$ |
| | scGPT | - | $0.4846_{\pm 0.0000}$ | $0.3016_{\pm 0.0000}$ | $0.3622_{\pm 0.0000}$ |
| | scFoundation | - | $0.4805_{\pm 0.0007}$ | $0.3110_{\pm 0.0008}$ | $0.3692_{\pm 0.0006}$ |
| | ALIGNED (MLP) | MLP only | $0.5360_{\pm 0.0019}$ | $0.3767_{\pm 0.0063}$ | $0.4192_{\pm 0.0062}$ |
| | | A | $0.5040_{\pm 0.0106}$ | $0.4307_{\pm 0.0197}$ | $0.4575_{\pm 0.0125}$ |
| | | A-R | $0.5252_{\pm 0.0042}$ | $0.6549_{\pm 0.0044}$ | $0.5697_{\pm 0.0024}$ |
| | | A-R-A | $0.5246_{\pm 0.0047}$ | $0.6528_{\pm 0.0064}$ | $0.5687_{\pm 0.0024}$ |
| | | A-R-A-R | $0.5248_{\pm 0.0046}$ | $0.6573_{\pm 0.0045}$ | $0.5698_{\pm 0.0028}$ |
| | ALIGNED (GNN) | GNN only | $0.5386_{\pm 0.0017}$ | $0.3820_{\pm 0.0109}$ | $0.4242_{\pm 0.0102}$ |
| | | A | $0.5154_{\pm 0.0163}$ | $0.4133_{\pm 0.0366}$ | $0.4447_{\pm 0.0260}$ |
| | | A-R | $0.5270_{\pm 0.0023}$ | $0.6555_{\pm 0.0021}$ | $0.5714_{\pm 0.0022}$ |
| | | A-R-A | $0.5261_{\pm 0.0032}$ | $0.6559_{\pm 0.0024}$ | $0.5711_{\pm 0.0025}$ |
| | | A-R-A-R | $0.5261_{\pm 0.0033}$ | $0.6596_{\pm 0.0019}$ | $0.5717_{\pm 0.0029}$ |
| Dixit | linear | - | $0.2827_{\pm 0.0000}$ | $0.1432_{\pm 0.0000}$ | $0.1807_{\pm 0.0000}$ |
| | GEARS | - | $0.4330_{\pm 0.0017}$ | $0.2704_{\pm 0.0029}$ | $0.3243_{\pm 0.0029}$ |
| | State | - | $0.1889_{\pm 0.0008}$ | $0.3530_{\pm 0.0010}$ | $0.1745_{\pm 0.0009}$ |
| | scGPT | - | $0.4361_{\pm 0.0001}$ | $0.2804_{\pm 0.0001}$ | $0.3335_{\pm 0.0001}$ |
| | scFoundation | - | $0.4335_{\pm 0.0004}$ | $0.2722_{\pm 0.0010}$ | $0.3260_{\pm 0.0009}$ |
| | ALIGNED (MLP) | MLP only | $0.4207_{\pm 0.0028}$ | $0.3930_{\pm 0.0184}$ | $0.4052_{\pm 0.0113}$ |
| | | A | $0.4126_{\pm 0.0022}$ | $0.4245_{\pm 0.0168}$ | $0.4173_{\pm 0.0069}$ |
| | | A-R | $0.4188_{\pm 0.0039}$ | $0.6591_{\pm 0.0036}$ | $0.4718_{\pm 0.0041}$ |
| | | A-R-A | $0.4181_{\pm 0.0036}$ | $0.6579_{\pm 0.0042}$ | $0.4711_{\pm 0.0038}$ |
| | | A-R-A-R | $0.4180_{\pm 0.0036}$ | $0.6630_{\pm 0.0014}$ | $0.4711_{\pm 0.0037}$ |
| | ALIGNED (GNN) | GNN only | $0.4509_{\pm 0.0068}$ | $0.3808_{\pm 0.0033}$ | $0.4069_{\pm 0.0040}$ |
| | | A | $0.4534_{\pm 0.0102}$ | $0.4147_{\pm 0.0251}$ | $0.4289_{\pm 0.0126}$ |
| | | A-R | $0.4585_{\pm 0.0048}$ | $0.6367_{\pm 0.0122}$ | $0.5082_{\pm 0.0046}$ |
| | | A-R-A | $0.4588_{\pm 0.0045}$ | $0.6376_{\pm 0.0130}$ | $0.5082_{\pm 0.0046}$ |
| | | A-R-A-R | $0.4589_{\pm 0.0047}$ | $0.6555_{\pm 0.0023}$ | $0.5110_{\pm 0.0047}$ |
| Adamson | linear | - | $0.2806_{\pm 0.0000}$ | $0.1866_{\pm 0.0000}$ | $0.2198_{\pm 0.0000}$ |
| | GEARS | - | $0.4687_{\pm 0.0018}$ | $0.3737_{\pm 0.0045}$ | $0.4132_{\pm 0.0024}$ |
| | State | - | $0.2868_{\pm 0.0017}$ | $0.3539_{\pm 0.0009}$ | $0.2339_{\pm 0.0013}$ |
| | scGPT | - | $0.4703_{\pm 0.0000}$ | $0.3563_{\pm 0.0001}$ | $0.4016_{\pm 0.0001}$ |
| | scFoundation | - | $0.4704_{\pm 0.0002}$ | $0.3678_{\pm 0.0013}$ | $0.4097_{\pm 0.0009}$ |
| | ALIGNED (MLP) | MLP only | $0.2888_{\pm 0.0030}$ | $0.3554_{\pm 0.0293}$ | $0.3128_{\pm 0.0077}$ |
| | | A | $0.4767_{\pm 0.0108}$ | $0.3725_{\pm 0.0315}$ | $0.4063_{\pm 0.0281}$ |
| | | A-R | $0.4682_{\pm 0.0102}$ | $0.4111_{\pm 0.0261}$ | $0.4325_{\pm 0.0179}$ |
| | | A-R-A | $0.4672_{\pm 0.0086}$ | $0.6467_{\pm 0.0056}$ | $0.5178_{\pm 0.0078}$ |
| | | A-R-A-R | $0.4674_{\pm 0.0078}$ | $0.6418_{\pm 0.0083}$ | $0.5178_{\pm 0.0077}$ |
| | ALIGNED (GNN) | GNN only | $0.5436_{\pm 0.0059}$ | $0.4319_{\pm 0.0317}$ | $0.4692_{\pm 0.0268}$ |
| | | A | $0.4925_{\pm 0.0450}$ | $0.5589_{\pm 0.1079}$ | $0.5023_{\pm 0.0226}$ |
| | | A-R | $0.5225_{\pm 0.0177}$ | $0.6321_{\pm 0.0097}$ | $0.5641_{\pm 0.0142}$ |
| | | A-R-A | $0.5205_{\pm 0.0168}$ | $0.6328_{\pm 0.0105}$ | $0.5611_{\pm 0.0127}$ |
| | | A-R-A-R | $0.5235_{\pm 0.0179}$ | $0.6504_{\pm 0.0026}$ | $0.5665_{\pm 0.0143}$ |

