# OpenReview forum: "Adaptive Data-Knowledge Alignment in Genetic Perturbation Prediction"
_ICLR.cc/2026/Conference — ICLR 2026 Poster_

### Official Review · Reviewer_YBGA · 2025-10-23

**Soundness:** 2
**Presentation:** 3
**Contribution:** 1
**Rating:** 4
**Confidence:** 4

**Summary:**

The paper presents ALIGNED, a method for predicting the effects of gene perturbations in single-cell transcriptomic data using prior gene regulatory networks (GRNs). Unlike earlier models that also leverage GRNs—such as GEARS—the proposed approach does not treat the regulatory information as static but instead refines it dynamically throughout training. The training process itself is complex, involving a multistage optimization scheme that combines gradient descent with a Monte Carlo, ultimately aiming to align predictions based on data with the ones based on GRN.

**Strengths:**

The authors employ a genuinely new strategy that leverages GRN for perturbation prediction that does not treat the GRN as static during training. They also devise a complex strategy of optimizing the neuro-symbolic alignment.

**Weaknesses:**

In terms of the accuracy of perturbation respnse predictions, the algorithm provides only marginal gains, while being notably resource-intensive—requiring up to 12 hours of training on an 80 GB GPU. The authors, nevertheless, argue that the true advantage of their method lies not in predictive accuracy but in its improved alignment with the prior GRN and in the refinement of the GRN during optimization. However, the first claim is problematic: the reported GRN-alignment metric seems circular, as the GRN serves both as an input to the model and as a reference for evaluation. The second claim—that the optimization refines the GRN—is also inconclusive, since the authors do not quantify the number or directionality of corrected regulatory links but instead report pathway enrichment scores, which appear only tangentially related.

**Questions:**

**I am willing to improve the score of the paper if:**

1. The authors explain why the knowledge consistency metric is important and prove that their way of computing it is not circular.
2. The authors conclusively show that their approach provides substantial benefits in terms of perturbation prediction as compared to other methods in the field, including the current state-of-the-art STATE.
3. The authors describe the optimization procedure from a practical point of view, explaining how different parameters at each stage influence the accuracy of predictions.
4. *(Minor)* The authors mention that two of the datasets were split randomly — please clarify what is meant by that.

---

> ### Author Response · Authors · 2025-11-26
> **Response to reviewer YBGA - Part 1/2**
>
> We appreciate your careful evaluation and insightful questions! We will address your concerns point by point.
>
> ---
>
> >**Q1: The authors explain why the knowledge consistency metric is important and prove that their way of computing it is not circular.**
>
> **A1 - Clarification on the knowledge consistency metric**
>
> Thank you for your insightful suggestions! Below, we address these pointwise.
>
> **(1.1) The importance of knowledge consistency**
>
> ALIGNED’s objective is not only to ensure predictive accuracy but to **adaptively integrate data-driven learning with GRN prior knowledge** and **to systematically perform knowledge refinement**. This requires evaluating how well predictions agree with mechanistic priors, which is **an aspect that standard predictive metrics cannot capture**.
>
> Knowledge consistency, therefore, serves as a **biological sanity check**: it ensures that model predictions remain grounded in known regulatory prior knowledge and do not drift toward biologically implausible or spurious correlations. This aligns with discussions in the literature highlighting that grounding models in a biological foundation is essential to avoid false discoveries and maintain interpretability [1,2,3].
>
> In addition, the knowledge-consistency metric directly supports biological interpretation. Identifying disagreements between data-driven predictions and prior knowledge is valuable because these inconsistencies can point to missing, outdated, or incorrect regulatory relationships in existing GRNs. For biologists, discrepancies provide actionable signals that can guide new experiments [4] or formulate clinical hypotheses [5,6].
>
> **(1.2) Clarification on circular evaluation of the knowledge consistency**
>
> We acknowledge that comparing predictions to the KB may appear circular. However, knowledge consistency is **not used as a stand-alone objective or a measurement of generalization** in the evaluation. Instead, it quantifies the extent to which the model predictions remain grounded in curated biological priors, analogous to checking whether outputs fall within a known domain distribution.
>
> The knowledge consistency metric forms only one part of the balanced consistency, which jointly evaluates agreement with both the KB and the data. Models that simply replicate the KB obtain perfect knowledge consistency but poor data consistency and therefore low balanced consistency, showing that **the metric does not reward KB memorization.**
>
> Importantly, we use the balanced consistency to evaluate if the adaptor in ALIGNED can make good trade-offs between inconsistent data and knowledge. When we train the adaptor, however, **it is never given the “correct” signal of the data-knowledge trade-off.** Thus, the metric evaluates whether predictions remain biologically coherent **without introducing circular information** into training or evaluation.
>
> Performance of KB memorization:
>
> Dataset | data cons. | knowledge cons. | balanced cons.
> --- | --- | --- | ---
> Norman et al. | 0.3216 | 1. | 0.4008
> Adamson et al. | 0.2859 | 1. | 0.3574
> Dixit et al. | 0.2891 | 1. | 0.3613
>
> ---
>
> **References**
>
> [1] I. Bendidi et al., ‘Benchmarking Transcriptomics Foundation Models for Perturbation Analysis : one PCA still rules them all’, Nov. 04, 2024, arXiv: arXiv:2410.13956. doi: 10.48550/arXiv.2410.13956.
>
> [2] K. Z. Kedzierska, L. Crawford, A. P. Amini, and A. X. Lu, ‘Zero-shot evaluation reveals limitations of single-cell foundation models’, Genome Biology, vol. 26, no. 1, p. 101, Apr. 2025, doi: 10.1186/s13059-025-03574-x.
>
> [3] J. W. Squair et al., ‘Confronting false discoveries in single-cell differential expression’, Nat Commun, vol. 12, no. 1, p. 5692, Sept. 2021, doi: 10.1038/s41467-021-25960-2.
>
> [4] R. J. Maizels, ‘A dynamical perspective: moving towards mechanism in single-cell transcriptomics’, Philosophical Transactions of the Royal Society B: Biological Sciences, vol. 379, no. 1900, p. 20230049, Mar. 2024, doi: 10.1098/rstb.2023.0049.
>
> [5] R. E. Baker, J.-M. Peña, J. Jayamohan, and A. Jérusalem, ‘Mechanistic models versus machine learning, a fight worth fighting for the biological community?’, Biology Letters, vol. 14, no. 5, p. 20170660, May 2018, doi: 10.1098/rsbl.2017.0660.
>
> [6] M. Chevalley, Y. H. Roohani, A. Mehrjou, J. Leskovec, and P. Schwab, ‘A large-scale benchmark for network inference from single-cell perturbation data’, Commun Biol, vol. 8, no. 1, p. 412, Mar. 2025, doi: 10.1038/s42003-025-07764-y.

---

> ### Author Response · Authors · 2025-11-26
> **Response to reviewer YBGA - Part 2/2**
>
> >**Q2: The authors conclusively show that their approach provides substantial benefits in terms of perturbation prediction as compared to other methods in the field, including the current state-of-the-art STATE.**
>
> **A2 - Comparison with STATE**
>
> We thank the reviewer for providing **the latest preprint SOTA** methods. We included STATE’s performance under the setting compatible with our protocol by training and evaluating on each dataset with held-out perturbations. We note, however, that STATE is primarily designed for cross-cell-context generalization, not prediction on unseen perturbations within the same cell line. Applying it in its intended mode would require training on perturbations from other cell lines, which would introduce the circularity issue similar to the note in Q1 (i.e., using information from the test perturbations to train STATE). For fairness, we therefore report STATE only under the non-circular, dataset-specific evaluation setting. **A complete comparison is included in Appendix F.6 of the revision.**
>
> Dataset | Method | Data cons. | KB cons. | Balanced cons.
> --- | --- | --- | --- | ---
> Norman  | STATE | .381 | .247 | .293
> Norman | ALIGNED (no refinement) | **.539** | .382 | .424
> Norman | ALIGNED (complete loop) | .526 | .660 | **.572**
> Adamson | STATE | .354 | .234 | .276
> Adamson | ALIGNED (no refinement) | .453 | .415 | .429
> Adamson | ALIGNED (complete loop) | **.459** | .656 | **.511**
> Dixit | STATE | .353 | .175 | .221
> Dixit | ALIGNED (no refinement) | .493 | .559 | .502
> Dixit | ALIGNED (complete loop) | **.524** | .650 | **.567**
>
> ---
>
> >**Q3: The authors do not quantify the number or directionality of corrected regulatory links but instead report pathway enrichment scores, which appear only tangentially related.**
>
> **A3 - Clarification of evaluation metrics in knowledge refinement**
>
> We thank the reviewer for this suggestion. In addition to pathway enrichment and topological analyses, we already quantify the corrected regulatory links directly using F1 scores, treating the original GRN as ground truth (Section 4.2). This captures both the number and directionality of recovered interactions.
>
> ---
>
> >**Q4: The authors describe the optimization procedure from a practical point of view, explaining how different parameters at each stage influence the accuracy of predictions.**
>
> **A4 - Hyperparameter selection**
>
> We thank you for your valuable questions and clarification requests regarding hyperparameters and ablations! **We included additional results and explanations in Appendix E.3, 4 of our revision**, and the observations are summarized as follows:
>
> - $k$ in the transitive closure of KB  is positively correlated with data consistency but negatively correlated with the knowledge consistency.
> - $\boldsymbol{w}$ in the adaptive neuro-symbolic alignment (Equation 4) is positively correlated with knowledge consistency but slightly compromises data consistency (predictive performance on test data).
> - $\theta$ in the adaptive alignment (Equation 4) has a minor effect on the performance.
> - $t_k$ and $t_0$ in the knowledge refinement are positively correlated with the performance, but improvements are insignificant when they are larger than the default values.
>
> ---
>
> >**Q5: (Minor) The authors mention that two of the datasets were split randomly — please clarify what is meant by that.**
>
> **A5 - Data split**
>
> We thank you for your suggestion to improve the clarity of our paper! All the experiments were run for 5 replicates on Adamson et al. and Dixit et al. We randomly split each of these perturbation datasets to create the train set and the unseen test set. Since the datasets contain only single perturbations, a random split would ensure no overlaps of perturbations between the train and the test set. On the Norman et al. dataset, a fixed split was selected to ensure all test perturbations are unseen. **We included details of the test split in the Appendix E.5.**
>
> ---
>
> >**Q6: while being notably resource-intensive—requiring up to 12 hours of training on an 80 GB GPU**
>
> **A6 - Computational cost**
>
> We greatly appreciate your comment regarding computational efficiency. In our experiments, we integrated multiple knowledge sources to predict perturbation responses for large gene sets (5,045 genes for the Norman et al. dataset, 5,060 genes for the Adamson et al. dataset and 5,012 genes for the Dixit et al. dataset). Importantly, ALIGNED not only operates at the genome-scale resolution, but also jointly performs (i) perturbation-response prediction and (ii) iterative knowledge refinement over the entire signed GRN. Under a single NVIDIA A100 (80 GB), we find ALIGNED training tractable, given the scale of the GRN and the knowledge refinement involved. However, we agree that improving training efficiency is a valuable direction, and consider this an exciting avenue for future work.

---

### Official Review · Reviewer_nsYx · 2025-10-28

**Soundness:** 2
**Presentation:** 2
**Contribution:** 2
**Rating:** 2
**Confidence:** 5

**Summary:**

This paper proposes to tackle a ternarized version of gene perturbation prediction through incorporating prior domain knowledge given by a knowledgebase in the learning process. They suggest to alternatingly refine the predicted outcomes and the knowledge base during the learning process and propose a gradient-free algorithm for this.

**Strengths:**

- The presented paper is original in that it presents a novel way of introducing prior knowledge into gene perturbation prediction models.

**Weaknesses:**

-	This paper considers a simplified ternarized version of the underlying problem of gene perturbation prediction, **which is not useful in practice**. The extent of change introduced by a prediction is essential to know. The de facto gold-standard challenge in the field right now is the virtual cell challenge held by the Arc institute (https://virtualcellchallenge.org/, [1]), for which the evaluation metrics already disclose that continuous predictions are necessary. I have seen academic works using this binary or ternary prediction framework to be able to use language models before, but that does not make the task in anyway useful.
-	There are **strong assumptions on how inhibitory and activating effects are interacting** that are not discussed in the main paper. I.e., to perform a query on the KB (the $\delta_{KB}$), it is assumed that activating and inhibitory effects are additive, which needs clear justification.
-	It is **unclear how scalable this approach is** in practice, the adjacency matrices are #genes x # genes, which can go up to n>60k (all annotated genes) or n>24k (protein-coding genes), in both cases resulting in restrictively large matrices. It is unclear from the writing what subset of genes is used in the experiments, but it appears to be a small selection which needs justification.
-	The writing is incomplete and in its current form the paper **can not be reproduced**.
1. How is the threshold $\theta$ defined and picked in practice? Why?
2. How is weight vector $w$ constructed exactly? A vector that “contains the number of training data samples[…] and the number of annotations from gene Ontology[…]” leaves a lot of questions – these are two numbers (counts) which clearly can not make up $w$. Do you sum per gene? Then add up the two numbers (KB and GO)? What is the rationale?
3. What are the number of genes (and which ones) considered for the experiments?
4. **What are the model architectures you use exactly**?
5. **Do you evaluate on hold-out data and KBs unseen during training**? What would the performance then be, e.g. evaluating on a PPI-derived KB or other datasets on the same (or even different) cell line, which is fairly common in the literature?
-	What is the comparison to SOTA methods, including the currently not discussed Morph [2], and STATE [3] models? Fig. 4,5 and Table 1 simply **miss a comparison of the actual prediction to any SOTA**.

[1] Rohaani, Y H et al. * Virtual Cell Challenge: Toward a Turing test for the virtual cell.* Cell 188(13), pages 3370-3374, 2025.

[2] He, C et al. * MORPH Predicts the Single-Cell Outcome of Genetic Perturbations Across Conditions and Data Modalities.* bioRxiv, DOI: https://doi.org/10.1101/2025.06.27.661992, 2025.

[3] Adduri, A K et al. * Predicting cellular responses to perturbation across diverse contexts with State.* bioRxiv, DOI:https://doi.org/10.1101/2025.06.26.661135, 2025.

**Questions:**

-	What is the justification on the assumption made for how activating and inhibitory effects propagate and interact?
-	Given that the KB is given as binary matrices, why can’t you do gradient-based optimization on this? You can follow a similar scheme as in 3.4 as far as the explanations go – what is prohibiting this?
-	What is the (exact!) architecture you are using in the experiments?
-	How many and which genes do you consider in the experiments? What happens if you consider more/less?
-	What are the precision and recall, given that the KB is so sparse, which is hidden in the F1?
-	What is the performance on training data?
-	What is the performance on unseen (hold-out) data?
-	What is the performance of KB reconstruction on an unseen KB (e.g. PPI-derived)?
-	What is the comparison to SOTA methods using common benchmarking protocols, both regarding metrics as well as separate data (see Weaknesses)?

---

> ### Author Response · Authors · 2025-11-26
> **Response to reviewer nsYx - Part 1/3**
>
> We thank the reviewer for their very detailed comments and careful evaluation. We appreciate the opportunity to strengthen several points in our paper in order to improve its clarity and presentation. Below, we clarify each point systematically.
>
> ---
>
> >**Q1: This paper considers a simplified ternarized version of the underlying problem of gene perturbation prediction, which is not useful in practice. The extent of change introduced by a prediction is essential to know. ... I have seen academic works using this binary or ternary prediction framework ... does not make the task in anyway useful.**
>
> **A1 - Clarifications on ternarized predictions.**
>
> We greatly appreciate the reviewer’s concern. We agree that continuous predictions are important in benchmarking settings such as the Virtual Cell Challenge. However, ALIGNED uses a ternary representation **not as a replacement for continuous modeling**, but because **our core objective is different: to adaptively integrate data-driven learning with GRN prior knowledge and to systematically perform knowledge refinement**.
>
> The ternarized representation **provides compatible resolution with downstream scientific applications** such as gene differential-expression analysis, pathway analysis, and ontology annotations, etc. [1,2,3]. Biological priors (activation, inhibition, no-effect) are conventionally expressed as discrete states [4], which become the practical interface for neuro-symbolic reasoning. The discrete abstraction also increases the interpretability and robustness of analysis, given the sparsity [5] and noise [6,7,8] in perturbation data.
>
> Owing to ALIGNED’s objective of identifying and resolving divergences between data-driven learning and prior knowledge, this discrete formulation provides the level of resolution needed to detect conflicts, propagate signals through neural and symbolic components, and refine the GRN in a transparent and interpretable way that biologists can leverage.
>
> ---
>
> >**Q2: There are strong assumptions on how inhibitory and activating effects are interacting that are not discussed in the main paper. I.e., to perform a query on the KB (the \delta_{KB}), it is assumed that activating and inhibitory effects are additive, which needs clear justification.**
>
> **A2 - Justifications for activation and inhibitory effect.**
>
> We greatly appreciate this suggestion! The literature [9, 10] emphasizes that it remains an open question whether additive or non-additive formulations better capture the true biological mechanisms underlying regulatory interactions. Since **we aim to ensure the computational tractability** of ALIGNED for KB query on a large scale, we **adopt the additive matrix operations in the implementation of KB query** $\delta_{\mathcal{KB}}$. We also discuss in the paper that richer, differentiable formulations are a natural direction for future work, whereas our current implementation serves as a practical and scalable solution.
>
> On the modelling perspective, however, **we do not assume that activating and inhibitory effects combine through simple additive rules**. Our GRN modelling follows the signed transitive-closure semantics defined in the bilinear recursive datalog program (Appendix B.1, Equation 7). This formulation is path-dependent and encodes how positive and negative regulations compose across multi-step pathways. While this abstraction simplifies the underlying biochemical kinetics, it is a widely used qualitative approach [11] to capture the direction of regulatory influence. **We have further highlighted these points in Section 2.2 and Appendix B.3 of our revision.**
>
> ---
>
> >**Q3: Given that the KB is given as binary matrices, why can’t you do gradient-based optimization on this? You can follow a similar scheme as in 3.4 as far as the explanations go – what is prohibiting this?**
>
> **A3 - Justification for the symbolic component.**
>
> We thank the reviewer for the thoughtful question! Our use of symbolic reasoning in Section 2.2 is motivated by the need to capture both direct and indirect regulatory effects in a principled way. The symbolic fixpoint computation (Equation 1) allows us to propagate regulatory influences through pathways of length up to k, thereby modeling multi-step interactions that cannot be represented by direct edges alone. While gradient-based optimization is indeed possible for direct binary matrices (an approach we adopt in Section 3.4), we need the symbolic fixpoint computation since **gradient-based approach does not by itself capture the recursive structure of indirect regulation**.

---

> ### Author Response · Authors · 2025-11-26
> **Response to reviewer nsYx - Part 2/3**
>
> >**Q4: It is unclear how scalable this approach is in practice, the adjacency matrices ... resulting in restrictively large matrices. It is unclear from the writing what subset of genes is used in the experiments ... which needs justification.**
>
> **A4 - Clarifications on scalability.**
>
> We thank the reviewer for raising this important point. In our experiments, the knowledge base integrates the OmniPath GRN with the GO-based interaction graph from Roohani et al., to predict perturbation response for 5,045 genes for the Norman et al. dataset, 5,060 genes for the Adamson et al. dataset and 5,012 genes for the Dixit et al. dataset, which follows the default setting in GEARS (Roohani et al.). We adopted the evaluation protocol (dataset settings and gene sets for response prediction) used in GEARS. These gene sets correspond to the full set of genes for which regulatory information is available in the combined knowledge base and are **not manually down-selected**.
>
> ---
>
> >**Q5: The writing is incomplete and in its current form the paper can not be reproduced.**
>
> **A5 - Clarifications on reproducibility.**
>
> We thank the reviewer for highlighting the perspective of reproducibility. While we aimed to facilitate reproducibility as best we could, due to space limitations, we included details of hyperparameter choices and detailed experimental results in the Appendix. To further accommodate reproducibility, **we now further explain the hyperparameter choices and experimental results in the Appendix Section E, F.**
>
> ---
>
> >**Q6: What is the (exact!) architecture you are using in the experiments? How many and which genes do you consider in the experiments? What happens if you consider more/less? What are the precision and recall, given that the KB is so sparse, which is hidden in the F1? What is the performance on training data? What is the performance on unseen (hold-out) data? What is the comparison to SOTA methods using common benchmarking protocols, both regarding metrics as well as separate data (see Weaknesses)?**
>
> **A6 -Pointwise responses.**
>
> We appreciate the reviewer’s thoughtful questions. Below, we address these related questions pointwise.
>
> **(6.1) Exact architecture**
>
> In our experiments, we compared two neural component architectures - MLP and GNN (Figure 3) to understand the effect of architectural choice on our framework’s performance. We explained the details of the neural component architectures in Section 4.1. From our experiments, GNN has shown slightly superior performance across multiple datasets.
>
> **(6.2) Precision and recall**
>
> Owing to the size of all performance statistics, **we include them in the Appendix F.6**.
>
> **(6.3) Performance on training data**
>
> Similar to the point above, since the size of all performance data is large, **we include all related results in the Appendix F.6**.
>
> **(6.4) Performance on unseen data and experiment protocol**
>
> All of our comparisons in Figure 3 are performed on unseen test split, and this is explained in Section 4.1. In Figure 3, we adopted the common benchmark protocol used in [9] to compare ALIGNED with GEARS scGPT scFoundation. Owing to the size of all performance statistics, we include the full results only in Appendix F.6. For reference and reproducibility, we include further specification of the dataset split in Appendix E.5.

---

> ### Author Response · Authors · 2025-11-26
> **Response to reviewer nsYx - Part 3/3**
>
> >**Q7: What is the comparison to SOTA methods, including the currently not discussed Morph [2], and STATE [3] models? Fig. 4,5 and Table 1 simply miss a comparison of the actual prediction to any SOTA.**
>
> **A7 - Comparison with STATE and clarification on SOTA comparisons**
>
> We thank the reviewer for providing **the latest preprint SOTA** methods. We included STATE’s performance under the setting compatible with our protocol by training and evaluating on each dataset with held-out perturbations. We note, however, that STATE is primarily designed for cross-cell-context generalization, not prediction on unseen perturbations within the same cell line. Applying it in its intended mode would require training on perturbations from other cell lines, which would use information from the test perturbations to train STATE. For fairness, we therefore report STATE only under the non-circular, dataset-specific evaluation setting. **We included the complete comparison results in Appendix F.6 of the revision.**
>
> Dataset | Method | Data cons. | KB cons. | Balanced cons.
> --- | --- | --- | --- | ---
> Norman  | STATE | .381 | .247 | .293
> Norman | ALIGNED (no refinement) | **.539** | .382 | .424
> Norman | ALIGNED (complete loop) | .526 | .660 | **.572**
> Adamson | STATE | .354 | .234 | .276
> Adamson | ALIGNED (no refinement) | .453 | .415 | .429
> Adamson | ALIGNED (complete loop) | **.459** | .656 | **.511**
> Dixit | STATE | .353 | .175 | .221
> Dixit | ALIGNED (no refinement) | .493 | .559 | .502
> Dixit | ALIGNED (complete loop) | **.524** | .650 | **.567**
>
>
> For Figure 4 and Table 1, we have clarified in Section 4.2 that, we haven't noticed existing methods that are suitable for comparison as far as our exploration.
>
> ---
>
> >**Q8: What is the performance of KB reconstruction on an unseen KB, e.g. PPI-derived?**
>
> **A8 -  Knowledge refinement cross-reference evaluation**
>
> Thank you for the helpful advice on referencing external KBs for the knowledge refinement! For the knowledge refinement experiment in Section 4.2 (Figure 4.a), we additionally compared the refined KB by ALIGNED to reference networks in CausalBench [11] (built from STRINGdb, CORUM, and ChipSeq). The non-sparse regularized baseline here ablated the knowledge refinement mechanism, replacing the L1 sparse regularization term with Frobenius regularization. **We included additional results and explanations in Appendix F.4**, showing that our method is able to learn interactions that are supported by external reference sources such as PPI or ChipSeq databases.
>
> Result on STRINGdb (F1 score of refined KB vs. CausalBench reference, full comparison in Appendix F.4):
> Noisy interactions | ALIGNED | non-sparse baseline
> --- | --- | ---
> 0%  | **.750** | .294
> 10% | **.681** | .316
> 30% | **.675** | .256
> 50% | .371 | .047
>
> ---
>
> **References**
>
> [1] Michael I Love, Wolfgang Huber, and Simon Anders. Moderated estimation of fold change and dispersion for rna-seq data with deseq2. Genome biology, 15(12):1–21, 2014. doi:10.1186/s13059-014-0550-8.
>
> [2] P. Badia-i-Mompel et al., ‘decoupleR: ensemble of computational methods to infer biological activities from omics data’, Bioinformatics Advances, vol. 2, no. 1, p. vbac016, Jan. 2022, doi: 10.1093/bioadv/vbac016.
>
> [3] P. Lo Surdo et al., ‘SIGNOR 4.0: the 2025 update with focus on phosphorylation data’, Nucleic Acids Res, p. gkaf1237, Nov. 2025, doi: 10.1093/nar/gkaf1237.
>
> [4] T. Schlitt and A. Brazma, ‘Current approaches to gene regulatory network modelling’, BMC Bioinformatics, vol. 8, no. 6, p. S9, Sept. 2007, doi: 10.1186/1471-2105-8-S6-S9.
>
> [5] E. Kernfeld, Y. Yang, J. S. Weinstock, A. Battle, and P. Cahan, ‘A systematic comparison of computational methods for expression forecasting’, Oct. 01, 2024, bioRxiv. doi: 10.1101/2023.07.28.551039.
>
> [6] R. Viñas Torné et al., ‘Systema: a framework for evaluating genetic perturbation response prediction beyond systematic variation’, Nat Biotechnol, pp. 1–10, Aug. 2025, doi: 10.1038/s41587-025-02777-8.
>
> [7] L. Li et al., ‘A Systematic Comparison of Single-Cell Perturbation Response Prediction Models’, Dec. 23, 2024, bioRxiv. doi: 10.1101/2024.12.23.630036.
>
> [8] S. Peidli et al., ‘scPerturb: harmonized single-cell perturbation data’, Nat Methods, vol. 21, no. 3, pp. 531–540, Mar. 2024, doi: 10.1038/s41592-023-02144-y.
>
> [9] C. Ahlmann-Eltze, W. Huber, and S. Anders, ‘Deep-learning-based gene perturbation effect prediction does not yet outperform simple linear baselines’, Nat Methods, pp. 1–5, Aug. 2025, doi: 10.1038/s41592-025-02772-6.
>
> [10] P. Badia-i-Mompel et al., ‘Comparison and evaluation of methods to infer gene regulatory networks from multimodal single-cell data’, Dec. 21, 2024, bioRxiv. doi: 10.1101/2024.12.20.629764.
>
> [11] M. Chevalley, Y. H. Roohani, A. Mehrjou, J. Leskovec, and P. Schwab, ‘A large-scale benchmark for network inference from single-cell perturbation data’, Commun Biol, vol. 8, no. 1, p. 412, Mar. 2025, doi: 10.1038/s42003-025-07764-y.

---

### Official Review · Reviewer_6cxn · 2025-10-30

**Soundness:** 3
**Presentation:** 3
**Contribution:** 3
**Rating:** 6
**Confidence:** 3

**Summary:**

This paper introduces ALIGNED, a neuro-symbolic learning framework that integrates gene regulatory knowledge bases (KBs) with data-driven learning to improve consistency between biological prior knowledge and empirical data. The core contribution is a bi-directional alignment mechanism:
Alignment stage — adapts the model to existing biological knowledge.
Refinement stage — updates and corrects the knowledge base itself using learned patterns.
It defines a novel Balanced Consistency metric to jointly evaluate data-knowledge coherence, addressing a gap in current hybrid bioinformatics methods that often treat biological priors as static and non-adaptive. The work is motivated by the high inconsistency (14–71%) between curated knowledge bases (OmniPath, GO, EcoCyc) and observed perturbation data, and proposes a continuous refinement cycle to resolve such mismatches.

**Strengths:**

Timely problem framing: Tackles a critical limitation in hybrid biological modeling—static priors that cannot evolve alongside new data.
Methodological novelty: The dual-stage alignment/refinement loop and Balanced Consistency metric provide a principled way to unify knowledge and data learning.
Strong empirical evidence: Across datasets (Norman et al. 2019, Adamson et al. 2016, Precise1K, E. coli), ALIGNED consistently outperforms baselines like GEARS, scGPT, and scFoundation in both prediction accuracy and knowledge consistency metrics.
Biological interpretability: Unlike pure neural models, the system supports symbolic reasoning and interpretable GRN (Gene Regulatory Network) updates.

**Weaknesses:**

Limited ablation of symbolic components: While gradient-free and gradient-based updates are both introduced, their relative contributions are only superficially analyzed.
Dataset diversity: Most evaluations center on transcriptional perturbation data; extension to morphological or cross-modal biological data is not shown.
Theoretical grounding: The abductive reasoning formulation (based on ABL) could benefit from clearer formal definitions and proofs of convergence or stability.
Potential overfitting to KB biases: Since refinement still depends on noisy or incomplete KBs, it’s unclear how ALIGNED avoids reinforcing existing errors.

**Questions:**

1. The paper introduces “bi-directional alignment” between the neural model and symbolic knowledge base. However, the exact mathematical formalization of the alignment operator and refinement update is somewhat ambiguous.
How is the “knowledge correction” signal computed and propagated?
Does the refinement step have convergence guarantees, or can it drift over repeated iterations?

2. The model includes multiple interacting components: gradient-based learning, abductive reasoning (gradient-free correction), and symbolic consistency scoring.
What is the contribution of each?
Would ALIGNED still outperform baselines without symbolic refinement?

3. The paper’s datasets are mostly transcriptomic (gene-expression–based). Could the same approach generalize to cross-modal or spatial data (e.g., imaging, proteomics)?
Does ALIGNED require structural priors specific to GRNs, or can it adapt to other biological or symbolic domains?

4. Baseline comparisons (GEARS, scGPT, etc.) are strong, but all are data-driven. What about existing neuro-symbolic or knowledge graph–augmented models (e.g., DeepProbLog, NeuroLogic, or KG-BERT)?

---

> ### Author Response · Authors · 2025-11-26
> **Response to reviewer 6cxn - Part 1/3**
>
> Thank you for recognizing the timeliness and novelty of our work. Your positive comments are a great motivation to improve our work! Below, we respond to each of your comments.
>
> ---
>
> >**Q1: The paper introduces ... However, the exact mathematical formalization of the alignment operator and refinement update is somewhat ambiguous. How is the “knowledge correction” signal computed and propagated? Does the refinement step have convergence guarantees, or can it drift over repeated iterations?**
>
> **A1 - Pointwise responses**
>
> We greatly appreciate your insightful questions! We address these questions one by one.
>
> **(1.1)  Mathematical formalization of alignment and refinement.**
>
> The mathematical formulation of adaptive alignment is described by Equation 4 and 5 in Section 3.3. Adaptive alignment jointly optimizes the neural component using standard cross-entropy loss and the adaptor using the discrete objective in Equation 4, where the adaptor selects, for each gene, whether to rely on the neural or symbolic prediction. Because Equation 4 defines a combinatorial, non-differentiable objective, we optimize the adaptor using REINFORCE, which provides a gradient estimator for stochastic discrete policies. **We added a clear formalization in Section 3.3 of the revision.**
>
> The knowledge refinement is mathematically formalized in Equation 6 in Section 3.4. This updates the entries of the KB via an approximation of non-differentiable Boolean matrix operations combined with L1-regularised proximal gradient descent, ensuring that only minimal adjustments are made to the GRN during refinement. **We included clearer details of the KB update after refinement in Section 3.4 the revision.**
>
> In our revision, we have explicitly highlighted adaptive alignment and knowledge refinement and their objective functions in Sections 3.3 and 3.4 to strengthen their connections.
>
> **(1.2) Convergence of REINFORCE.**
>
> REINFORCE is known to have convergence guarantees [1] under standard assumptions when the learning rate follows the Robbins–Monro conditions [2,3]. In practice, as in many applications of REINFORCE [4], we use a constant learning rate, which is empirically stable for the adaptor module in our setting.
>
> **(1.3) Convergence of align-refine loop.**
>
> We clarify that the convergence of the align-refine loop is guaranteed under theoretical assumptions that: (a) the neural prediction       $\hat{\boldsymbol{Y}}$ aligns better with $\bar{\boldsymbol{Y}}$ after update, and (b) the symbolic prediction $\delta_\mathcal{KB}(\boldsymbol{X})$ aligns better with $\bar{\boldsymbol{Y}}$ after the knowledge refinement. The inconsistency measure defined in Equation 2 is non-negative (lower-bounded by 0). It is jointly reduced by both the alignment and refinement steps since each iteration updates the neural component, the adaptor, and the symbolic component using the same integrated predictions. Thus, across iterations, the inconsistency decreases monotonically until it reaches a fixed point where further updates to either module no longer reduce disagreement. **We included a discussion in Appendix Section C.**
>
> In practice, ALIGNED converges without the theoretical assumptions holding strictly. Empirically, we observe that the loop stabilizes within a small number of iterations (typically 2-3), which is demonstrated by Figure 5.
>
> ---
>
> **References**
>
> [1] R. S. Sutton and A. G. Barto, Reinforcement Learning: An Introduction, 2nd edn. in Adaptive Computation and Machine Learning series. Cambridge, MA, USA: MIT Press, 2018.
>
> [2] H. Robbins and S. Monro, ‘A Stochastic Approximation Method’, The Annals of Mathematical Statistics, vol. 22, no. 3, pp. 400–407, Sept. 1951, doi: 10.1214/aoms/1177729586.
>
> [3] Richard S. Sutton, David A. McAllester, Satinder P. Singh, Yishay Mansour, ‘Policy Gradient Methods for Reinforcement Learning with Function Approximation’, NeurIPS 1999.
>
> [4] Peter Henderson, Riashat Islam, Philip Bachman, Joelle Pineau, Doina Precup, David Meger, ‘Deep Reinforcement Learning That Matters’, AAAI 2018, doi: https://doi.org/10.1609/aaai.v32i1.11694.

---

> ### Author Response · Authors · 2025-11-26
> **Response to reviewer 6cxn - Part 2/3**
>
> >**Q2: The model includes multiple interacting components: gradient-based learning, abductive reasoning (gradient-free correction), and symbolic consistency scoring. What is the contribution of each? Would ALIGNED still outperform baselines without symbolic refinement?**
>
> **A2 - Contribution of each component**
>
> Thank you for highlighting this important point. In Figure 5, we compare three progressively richer variants:
>
> 1) Neural-only model with no symbolic refinement (baseline) - a GNN trained solely on labelled data, with no symbolic alignment or refinement.
> 2) Adaptive neuro-symbolic alignment without refinement (“A”) - adds the alignment procedure on top of the neural baseline, but without updating the knowledge base.
> 3) Complete ALIGNED framework (“R”) - includes both adaptive alignment and the gradient-based symbolic refinement steps.
>
> These ablations isolate the contribution of each component. We observe that:
> - **Adaptive alignment** alone improves knowledge consistency (“symbolic consistency score” in the comment), showing that the adaptor effectively incorporates information from the symbolic component.
> - **Symbolic refinement** provides a substantial additional improvement in knowledge consistency, while maintaining accuracy comparable to the baseline. This demonstrates that refinement is essential for resolving inconsistencies between the neural predications and the GRN and enables ALIGNED to achieve its highest balanced consistency.
>
> In Figure 3, the ALIGNED framework with GNN and MLP performed adaptive neuro-symbolic alignment, but not refinement. This shows that the adaptor alone enables ALIGNED to outperform existing baselines by effectively using prior knowledge. **We have clarified these results and added further details in the revised Ablations Section (Section 4.4).**
>
> ---
>
> >**Q3: The paper’s datasets are mostly transcriptomic (gene-expression–based). Could the same approach generalize to cross-modal or spatial data (e.g., imaging, proteomics)? Does ALIGNED require structural priors specific to GRNs, or can it adapt to other biological or symbolic domains?**
>
> **A3 - Cross-modal generalization**
>
> We thank the reviewers for recognising the potential impact of ALIGNED on other data modalities. In the current paper, we focus specifically on single-cell transcriptomic perturbation data, as this is where large-scale perturbation datasets and GRN priors are more mature.
>
> Conceptually, ALIGNED does not rely on symbolic operations unique to gene regulatory networks. The framework only assumes the availability of (i) a neural predictor, (ii) a knowledge base that provides a symbolic consistency measurement, and (iii) an alignment/refinement loop that reconciles them. In principle, the symbolic module could encode other forms of structured biological priors-such as protein–protein interaction, metabolic pathways or spatial neighbourhood graphs. While we have not yet evaluated ALIGNED on cross-modal or spatial datasets, we are encouraged by these suggestions to explore extensions in future work that leverage structural priors. **Potential future works are further discussed in Section 6 of the revision.**

---

> ### Author Response · Authors · 2025-11-26
> **Response to reviewer 6cxn - Part 3/3**
>
> >**Q4: Baselines ... What about existing neuro-symbolic or knowledge graph–augmented models (e.g., DeepProbLog, NeuroLogic, or KG-BERT)?**
>
> **A4 - Symbolic or mechanistic baselines**
>
> Thank you for this helpful suggestion. We would like to clarify that since GEARS (compared in Section 4.1) uses a Gene Ontology-derived knowledge graph to train its gene perturbation embeddings, we consider GEARS to be a knowledge graph augmented approach. Meanwhile, other existing neuro-symbolic systems (such as DeepProbLog or NeuroLogic) or knowledge-graph augmented frameworks (such as KG-BERT) are designed for tasks fundamentally different from perturbation response prediction. These methods operate on probabilistic logical queries, constrained text generation, or knowledge graph link prediction, and they do not support (i) genome-scale regulatory effect modelling, (ii) high-dimensional biological response prediction, or (iii) iterative refinement of a large symbolic structure such as a GRN. As a result, we do not consider them directly applicable to our setting nor meaningful baselines for this task.
>
> That said, we agree with the reviewer that comparisons to mechanistic or prior-aware models are valuable. We have evaluated a recent mechanistic GRN-based perturbation simulator published in Sep. 2025 [5] on the Adamson et al. CRISPRi dataset. This model is the closest and latest existing approach that performs symbolic reasoning or mechanistic simulation over a curated GRN. Since we only examined under the default setting due to its substantial hyperparameter tuning requirements and the time constraints, its performance in our evaluation is not strong enough as a baseline.
>
> Method | Dataset | Data Cons. | Knowledge Cons. | Balanced Cons.
> --- | --- | --- | --- | ---
> Aguirre et al. [5] | Adamson et al. | 0.2808 | 0.2800 | 0.2804
>
> ---
>
> >**Q5: Theoretical grounding: The abductive reasoning formulation (based on ABL) could benefit from clearer formal definitions and proofs of convergence or stability.**
>
> **A5 - Updates in formalization**
>
> We thank the reviewer for noting the unclear formalizations. **A clearer formal definition of ABL is provided in Section 2.3 of our revision.** A discussion of convergence is included in the revision (Appendix C), and we refer the reviewer to our response (1.3).
>
> ---
>
> >**Q6: Potential overfitting to KB biases: Since refinement still depends on noisy or incomplete KBs, it’s unclear how ALIGNED avoids reinforcing existing errors.**
>
> **A6 - Clarifications on situations when KB is noisy**
>
> In Section 4.2, we corrupted the OmniPath KB with random noise and then applied ALIGNED’s refinement using synthetic data from the clean KB. As Figure 4 shows, the method successfully recovered the original structure (measured in accuracy - Figure 4a, topology- Figure 4b, and pathway enrichment - Figure 4c), demonstrating that refinement mitigates rather than reinforces KB errors.
>
> Importantly, ALIGNED also adapts its reliance on the KB based on reliability of knowledge. When symbolic knowledge is relatively less reliable, the adaptor uses more neural predictions during refinement. To support this, we conducted an experiment to stratify edges in the OmniPath knowledge base by curation effort as a proxy for reliability and evaluated how often the adaptor selects neural predictions for refinement. **We have included the results and further explanations in Appendix F.1.** Results show that the framework automatically downweights symbolic predictions when the GRN is less reliable, and therefore less susceptible to noise in the knowledge base.
>
> Curation effort | KB usage | data usage
> --- | --- | ---
> 0      | 13.3% | 86.7%
> [1,5)  | 24.7% | 75.3%
> [5,10) | **31.9%** | 68.1%
> \>=10  | **31.7%** | 68.3%
>
> We further cross-referenced OmniPath with external databases (**full results in Appendix F.2**). Again, ALIGNED relied mainly on neural predictions when evidence was sparse or inconsistent, reducing the risk of propagating noisy symbolic signals.
>
> Results on STRINGdb
> Edges |KB usage | data usage
> --- | --- | ---
> sparse | 10.5% | **89.5%**
> high confidence | **74.6%** | 25.4%
> low confidence | 54.8% | 45.2%
>
> ---
>
> **Reference**
>
> [5] M. Aguirre, J. P. Spence, G. Sella, and J. K. Pritchard, ‘Gene regulatory network structure informs the distribution of perturbation effects’, PLOS Computational Biology, vol. 21, no. 9, p. e1013387, Sept. 2025, doi: 10.1371/journal.pcbi.1013387.

---

### Official Review · Reviewer_Zesh · 2025-11-09

**Soundness:** 3
**Presentation:** 2
**Contribution:** 3
**Rating:** 6
**Confidence:** 3

**Summary:**

This paper proposes ALIGNED, a neuro-symbolic framework that jointly optimizes a neural predictor and a symbolic reasoning module over GRNs via abductive learning. The method introduces (i) a balanced consistency metric that merges agreement with data and with knowledge bases, (ii) an adaptive alignment mechanism (learned with REINFORCE) to select per-gene whether to trust neural or symbolic predictions, and (iii) gradient-based knowledge refinement with sparse regularization to update GRNs. Experiments on Norman, Dixit, Adamson (human/mouse) and an E. coli setting show higher “balanced consistency” than recent baselines and indicate that refinement can re-discover biologically meaningful relations.

**Strengths:**

Overall, I think this paper has a potentially significant contribution to perturbation-response prediction and neuro-symbolic learning, with a thoughtful framing of data-vs-knowledge trade-offs. With clarifications on theoretical assumptions, ablation coverage, and computational costs, I’d be inclined to raise my score.

**Weaknesses:**

1. The paper’s core ideas: balanced consistency, per-output adaptive selection, and sparse refinement—are compelling, but please delineate what is novel vs. adapted from prior ABL/REINFORCE literature (e.g., ABL variants, learnable trade-offs). A table that maps each ALIGNED component to closest prior art and states the delta would help.

2. Results implicitly require that either the neural model can approximate the true response mapping or the KB’s transitive closure (up to path length k) can approximate symbolic ground truth. Please state these assumptions explicitly and discuss failure modes when (a) GRN coverage is sparse/bias-prone or (b) δ_{KB} is systematically wrong for some modules. What guarantees (if any) are possible for convergence of the alternating align/refine loop under misspecification?

**Questions:**

1. How are threshold θ and weights w tuned—per dataset or globally? Any risk of overfitting via these hyperparameters?
2. For double-vs-single perturbations, does k need to scale, and does refinement overfit to short-cycle artifacts?

Minor comments / nits
1. Typos/formatting: “Algin 2” → Align 2; “ALIGEND” → ALIGNED; ensure consistent “knowledge base (KB)” capitalization.
2. In Fig. 3–5 captions, explicitly state metric definitions (macro vs micro F1).

---

> ### Author Response · Authors · 2025-11-26
> **Response to reviewer Zesh - Part 1/4**
>
> We greatly appreciate your positive feedback regarding the novelty and contribution of our work, which is a great encouragement for us to continue improving our work! Below, we will address your concerns point by point.
>
> ---
>
> >**Q1: Please delineate what is novel vs. adapted from prior ABL/REINFORCE literature (e.g., ABL variants, learnable trade-offs). A table that maps each ALIGNED component to closest prior art and states the delta would help.**
>
> **A1 - Comparison of ALIGNED components**
>
> Thank you for finding the core ideas of our work compelling! As requested, we provide a table below that maps each component of ALIGNED to its most related work in the literature and highlights what is new in our framework.
>
> -- | Neural component | Symbolic component | Adaptor | Bidirectional update
> --- | --- | --- | --- | ---
> **Prior work** | **GNN / MLP** | **BMLP** [1]: Provides highly efficient symbolic modeling of large-scale biological networks. Enables symbolic reasoning on high-performance hardware | **REINFORCE** [2]: Supports gradient-free joint optimization with neural networks and learnable optimization policies;  **ABL_refl** [3]: Provides an end-to-end framework for data-knowledge integration at the prediction level, supporting joint optimization of gradient-free policy and neural training. | **Basic ABL** [4]: Only updates the neural component, views knowledge bases as ground truth constraints; **ABL_nc** [5]: (1) Enables knowledge update at the concept level;  (2) Only updates the knowledge domain at the first iteration, and has limited scalability.
> **ALIGNED** | Same as prior work | (1) Extends the existing BMLP framework with bi-linear programming for modelling non-monotonic regulatory effects; (2) Uses BMLP to express ternary regulatory effects, instead of only binary activation/inhibition. | (1) Views neural and symbolic components as imperfect information sources. Makes balanced trade-offs, aims for more reliable prediction and knowledge refinement. (2)  Enables domain-specific optimization objective (by using w) for biological applications. |(1) Views knowledge bases as dynamic, instead of as static constraints; (2) Enables large-scale, multi-round knowledge refinement at the relation level.
>
> To summarise, ALIGNED advances from prior abductive learning (ABL) and neuro-symbolic approaches, in that it does **not assume the knowledge base to be fixed or absolutely correct**. Instead, ALIGNED **adaptively aligns** neural and symbolic predictions to leverage the most reliable signals and **actively refines** incomplete or noisy knowledge bases during training. **We have further clarified this point in Section 5 in the revision**.
>
> ---
>
> **References**
>
> [1] Lun Ai. Boolean matrix logic programming on the gpu, 2025. URL https://arxiv.org/abs/2408.10369.
>
> [2] Ronald J. Williams. Simple statistical gradient-following algorithms for connectionist reinforcement learning. Machine Learning, 8(3):229–256, May 1992. ISSN 1573-0565. doi:10.1007/BF00992696. URL https://doi.org/10.1007/BF00992696.
>
> [3] Wen-Chao Hu, Wang-Zhou Dai, Yuan Jiang, and Zhi-Hua Zhou. Efficient rectification of neuro-symbolic reasoning inconsistencies by abductive reflection. In Proceedings of the AAAI Conference on Artificial Intelligence, volume 39, pp. 17333–17341, 2025.
>
> [4] Zhi-Hua Zhou. Abductive learning: towards bridging machine learning and logical reasoning. Science China Information Sciences, 62(7):76101, 2019.
>
> [5] Yu-Xuan Huang, Wang-Zhou Dai, Yuan Jiang, and Zhi-Hua Zhou. Enabling knowledge refinement upon new concepts in abductive learning. In Proceedings of the AAAI Conference on Artificial Intelligence, volume 37, pp. 7928–7935, 2023.

---

> ### Author Response · Authors · 2025-11-26
> **Response to reviewer Zesh - Part 2/4**
>
> >**Q2: Results implicitly require that either the neural model ... or the KB ... can approximate ground truth. Please state these assumptions explicitly and discuss failure modes when (a) GRN coverage is sparse/bias-prone or (b) δ_{KB} is systematically wrong for some modules. What guarantees (if any) are possible for convergence of the alternating align/refine loop under misspecification?**
>
> **A2 - Pointwise responses**
>
> Thank you for these insightful comments and constructive suggestions to help us improve our paper!
>
> **(2.1) KB / data assumptions**
>
> We agree with the reviewer’s high-level intuition that ALIGNED leverages whichever source, data, or prior knowledge provides the more reliable signal. Owing to the imperfections in both single-cell sequencing data and prior knowledge, we would like to note that it cannot be effectively determined if the neural or symbolic component actually approximates the true perturbation response. Therefore, ALIGNED requires only a weaker assumption than the formulation suggested in the comment.
>
> **In our revision (Section 3.2), we explicitly clarify our assumption**: (i) when perturbation data is relatively more reliable than prior knowledge, the adaptor would increase reliance on the neural component, (ii) when the prior knowledge provides more reliable information, the contribution of the symbolic component to overall predictions would be higher.
>
> To support this assumption, we conducted an experiment to stratify edges in OmniPath by their curation effort (a proxy for knowledge reliability) and measured how often the adaptor selects the symbolic versus neural predictions in each regime. ALIGNED did not have access to this curation effort information during training.
>
> Our results show that the adaptor automatically down-weights the symbolic predictions when the GRN is less reliable (low curation effort) and relies more heavily on the neural predictor in those cases. Conversely, when the KB edges are better curated, the adaptor increases its symbolic usage. **Results are included in Appendix Section F.1**.
>
> Curation effort | KB usage | data usage
> --- | --- | ---
> 0      | 13.3% | 86.7%
> [1,5)  | 24.7% | 75.3%
> [5,10) | **31.9%** | 68.1%
> \>=10  | **31.7%** | 68.3%
>
> **(2.2) GRN failure modes**
>
> We conducted a new experiment using multiple external knowledge sources, STRINGdb (PPI), CORUM (PPI), Chip-Atlas (ChipSeq) and ENCODE (ChipSeq), as independent cross-references. We stratified gene pairs into three categories: (i) not regulated in both (sparse), (ii) regulated in both (high confidence), (iii) regulated only in KB or reference (low confidence). The rationale is that interactions supported by both our KB (OmniPath) and an external reference are more likely to be reliable, whereas interactions found in only one source may imply biases, incompleteness, or systematic errors.
>
> The results demonstrate that when coverage is sparse or inconsistent, ALIGNED relies heavily on the neural component (~90%). This prevents the model from propagating uncertain symbolic signals. In addition, when prior knowledge is less reliable, the symbolic component contributes less to the overall predictions compared to high-confidence edges. **We included these results in Appendix Section F.2**.
>
> Results on STRINGdb (full results in Appendix F.2):
> Edges |KB usage | data usage
> --- | --- | ---
> sparse | 10.5% | **89.5%**
> high confidence | **74.6%** | 25.4%
> low confidence | 54.8% | 45.2%
>
> **(2.3) Convergence guarantees under misspecification**
>
> We demonstrate empirically whether ALIGNED converges when KB contains misspecification or noise, and its performance. We randomly sampled interactions (3 repeats) that both exist in PPI (STRING or Corum), ChipSeq and OmniPath (the knowledge base we’re using), which can have a high confidence to be true interactions, and removed them from the OmniPath. We ran ALIGNED using the KB with removed edges to examine whether the removed interactions can be recovered. Results show ALIGNED converged, and we observed a high recovery rate of removed confident edges upon convergence. **We have included the results in Appendix F.3.**
>
> Replicate | Recovered interactions / Total
> --- | ---
> 1 | 11 / 12
> 2 | 10 / 10
> 3 | 10 / 11

---

> ### Author Response · Authors · 2025-11-26
> **Response to reviewer Zesh - Part 3/4**
>
> >**Q3: How are threshold θ and weights w tuned—per dataset or globally? Any risk of overfitting via these hyperparameters? For double-vs-single perturbations, does k need to scale, and does refinement overfit to short-cycle artifacts?**
>
> **A3 - Hyperparameters and ablations**
>
> We thank you for your valuable questions and clarification requests regarding technical details! We address these questions pointwise.
>
> **(3.1) Hyperparameter $\boldsymbol{w}$.**
>
> The weight vector $\boldsymbol{w}$ is computed separately for each dataset, but always using the same fixed procedure described in Appendix E.2. In particular, the component of $\boldsymbol{w}$ that reflects data-knowledge inconsistency is **derived only from the training split**, while the other components encode priors from the structured knowledge bases. This makes $\boldsymbol{w}$ a dataset-specific but **non-hand-tuned quantity**. Importantly, model generalization was assessed on held-out test data (Figure 3), where ALIGNED achieved significantly higher knowledge consistency and slightly higher data consistency compared to all baselines, indicating that the hyperparameter choices do not lead to overfitting. **We have further explained the setting of w in Appendix E.2. We also included additional results in Appendix E.3 of our revision** to show that every term in $\boldsymbol{w}$’s formula is positively correlated with knowledge consistency but slightly compromises data consistency.
>
> **(3.2) Hyperparameter $\theta$.**
>
> The hyperparameter $\theta$ was set globally across all datasets. **We have included additional results in Appendix E.3** to demonstrate that theta does not have a significant effect on the performance.
>
> **(3.3) Selection of $k$.**
>
> In our experiments, $k$ was not varied across datasets and single / double perturbations. For either single or double perturbations, $k=4$ covers more than 90% interactions of the transitive closure in practice. **We have included additional results in Appendix E.3** to demonstrate that $k$ is positively correlated with data consistency but negatively correlated with the knowledge consistency.
>
> **(3.4) Refinement does not overfit to short-cycle artifacts.**
>
> Our refinement mechanism uses sparse (L1) regularization, which explicitly discourages unnecessary additions to the GRN and helps prevent overfitting to shortcut or indirect regulatory effects. To examine whether refinement introduces short-cycle artifacts, we extended the GRN-refinement experiment in Section 4.2 by comparing the proportion of direct versus indirect interactions recovered after refinement. **The results are included in Appendix F.5 in the revision**.
>
> In this analysis, we treat the original GRN as the ground-truth interaction structure and evaluate how many direct and indirect edges are reconstructed after introducing noise. The results show that ALIGNED's refinement predominantly recovers direct interactions, while indirect ones remain limited. In contrast, the non-sparse baseline (using Frobenius regularization) overfits to indirect interactions, indicating that it is more prone to learning shortcut or small-cycle artifacts. These findings suggest that our sparse refinement strategy effectively guards against overfitting to short-cycle artifacts.
>
> Noisy edges | ALIGNED: direct | ALIGNED: indirect | non-sparse: direct | non-sparse: indirect
> --- | --- | --- | --- | ---
> 0%  |  **81.4%** |  17.0% |  16.6% |  68.0%
> 10% |  **71.8%** |  23.2% |  24.1% |  53.6%
> 30% |  **71.2%** |  18.8% |  11.9% |  70.5%
> 50% |  **58.6%** |  24.7% |  0.6% |  36.8%

---

> ### Author Response · Authors · 2025-11-26
> **Response to reviewer Zesh - Part 4/4**
>
> >**Q4: With clarifications on theoretical assumptions, ablation coverage, and computational costs, I’d be inclined to raise my score.**
>
> **A4 - Pointwise responses**
>
> Thank you for your valuable suggestions! Below, we address your comments pointwise.
>
> **(4.1) Theoretical assumptions.** We have provided an extensive response in our answer A2 covering our theoretical assumptions.
>
> **(4.2) Ablations.** We have further **clarified the details of ablation in Section 4.4 of the revision**. We examined different ALIGNED components, including (i) neural component-only, (ii) adaptive neuro-symbolic alignment without refinement, (iii) the complete ALIGNED framework. We also performed ablations on KB and the adaptor loss (Equation 4) using the Norman et al. dataset and a GNN as the neural component.
>
> **(4.3) Computational cost.**
>
> We greatly appreciate your suggestion regarding computational cost. We have made these experimental details and runtime statistics more prominent in our revision for clarity.
> All experiments were conducted using a single Intel Xeon Gold 6342 CPU (2.80 GHz, 32 GB memory) and a single NVIDIA A100 GPU (80 GB). Our knowledge base integrates the Omnipath GRN and the GO-based gene interaction graph from Roohani et al., to predict perturbation responses for large gene sets (5,045 genes for the Norman et al. dataset, 5,060 genes for the Adamson et al. dataset and 5,012 genes for the Dixit et al. dataset). Under this setting, training ALIGNED requires approximately 10–12 hours per run. The majority of the computation time was spent on the adaptive alignment and knowledge base refinement steps. These components include **gradient-free optimization procedures**, which are inherently more computationally intensive than standard gradient-based neural updates.
>
> Under the current experimental setup, the training cost of ALIGNED is manageable given the size of the GRN and the complexity of the refinement procedure. That said, we agree that further improving computational efficiency would be highly valuable, and we view this as an important direction for future development.
>
> ---
>
> >**Q5: Minor comments / nits: 1. Typos/formatting: “Algin 2” → Align 2; “ALIGEND” → ALIGNED; ensure consistent “knowledge base (KB)” capitalization. 2. In Fig. 3–5 captions, explicitly state metric definitions (macro vs micro F1).**
>
> **A5 - Pointwise responses**
>
> **(5.1) Typos and formatting mistakes.** Thank you very much for mentioning these issues! We have updated them in the revision.
>
> **(5.2) Metric definition.** We thank you for pointing these out. We have clarified that all F1 scores are macro in the revision.

---

### Author Response · Authors · 2025-11-26
**Quick Reference of Revisions**

**Quick Reference of Reviewer Comments and Corresponding Revisions**

We denote S. for *Strengths*, W. for *Weaknesses*, and Q. for *Questions* here.

**Response to Reviewer Zesh**
Review comment | Concern | Our response
--- | --- | ---
S.1  | Theoretical assumptions of the adaptor | Clarified in Section 3.2; added additional experiments in the Appendix F.
S.2  | Ablations | Summarized existing ablations in Section 4.4 and added new ablations in the Appendix E.
S.3  | Computational cost | Clarified in Section 4.
W.1 | Novelty comparison | Added table in response A1; clarified in Section 5.
Q.1 | Hyperparameter selection | Explained and added experiments in the Appendix E.
Q.2 | Overfitting to GRN biases and shortcuts | Explained in response A2.3 and A3.4; Conducted extended experiments; added analysis in Appendix F.
Q.3 | Typos and unclear caption | Updated in the main text.

---

**Response to Reviewer 6cxn**
Review comment | Concern | Our response
--- | --- | ---
W.1 & Q.1 | Mathematical formalization and convergence | Clarified in Section 3.3; added convergence discussion in Appendix C.
W.2 & Q.2 | Ablations | Summarized existing ablations in Section 4.4; added new ablations Appendix E.
W.3 & Q.3 | Cross-modality generalization | Clarified that ALIGNED can be applied to other structured priors; added discussion to Section 6.
W.4 & Q.4 | Missing neuro-symbolic baselines | Explained the suggested baselines are not applicable; added a mechanistic baseline in response A4.

---

**Response to Reviewer nsYx**
Review comment | Concern | Our response
--- | --- | ---
W.1 | Usefulness of ternary predictions | Clarified usefulness and necessity in response A1.
W.2 & Q. 1 | Assumptions on inhibitory and activating effects | Clarified in response A2, Section 2.2 and Appendix B.
Q.2 | Purpose of symbolic reasoning | Clarified need for modelling indirect interactions in response A3.
W.3 | Scalability and gene selection | Clarified ALIGNED operates on genome-scale and is used with no manual down-selection in A4.
W.4 & Q.5-7 | Reproducibility concerns | Clarified implementation details in response A6; Expanded hyperparameter in Appendix E; Added detailed metrics and experimental specifications in Appendix E, F.
W.5 & Q.9 | Comparison to STATE or MORPH | Added results to Section 4.1, Appendix F.6 and response A7.
Q.8 | Refinement on unseen KB | Added cross-reference evaluation in response A8 and Appendix F.

---

**Response to Reviewer YGBA**
Review comment | Concern | Our response
--- | --- | ---
W.1 | Computational cost | Clarified in Section 4.
W.2 & Q. 1 | Importance and circularity of proposed metrics | Clarified importance and purpose of metrics in response A1.
Q.2 | Comparison to STATE | Added results to Section 4.1, Appendix F.6 and response A2.
W.3 & Q. 3 | Quantifying corrected GRN links | Clarified this analysis was included already.
Q.3 | Details of optimization procedure | Clarified details of hyperparameters and added experiments in Appendix E.
Q.4 | Clarification on random splits | Clarified details in Appendix E.

---

### Author Response · Authors · 2025-12-01
**Summary of response to concerns**

**Summary of Responses to Key Reviewer Concerns**

We provide below a concise summary of how our rebuttal addresses the key concerns raised:

**1. Overall contribution** (Response to reviewer nsYx and YGBA). We thank reviewers nsYx and YGBA for highlighting their focus on predictive accuracy. Our contributions address important gaps in current perturbation-response modeling that extend beyond purely accuracy-based objectives:
- **(i)** predictions that are **better grounded on established biological knowledge** through a novel method in incorporating prior knowledge;
- **(ii) Continuous evolution of biological knowledge** based on both data-driven perturbation predictions and existing curated knowledge;
- **(iii)** Biologically interpretable outputs that are **compatible with downstream biological research applications**.

Across datasets, ALIGNED **exceeds or matches SOTA methods in predictive accuracy** while providing these **additional benefits not offered by existing approaches**.

**2. Ternarized prediction is necessary and useful** (Response to reviewer nsYx - Q1). The ternary representation is required to provide an appropriate resolution for aligning data-driven learning with biological priors but also remains meaningful for downstream scientific interpretation.

**3. The knowledge consistency metric is essential for biologically grounded prediction** (Response to reviewer YBGA - Q1). Standard predictive metrics cannot evaluate whether predictions adhere to mechanistic regulatory logic. The knowledge consistency metric captures this dimension and supports biologically interpretable outputs.

**4. The consistency evaluation is not circular** (Response to reviewer YBGA - Q1). Knowledge consistency is only one component of the balanced consistency metric, which does not reward circularity as we demonstrated, but evaluates the data-knowledge trade-off. Importantly, ALIGNED is never given the “correct” signal of the trade-off.

**5. ALIGNED is scalable** (Response to reviewer nsYx - A4/YBGA - Q6). ALIGNED predicts perturbation responses for the full set of genes in each dataset (>5,000 genes). Crucially, ALIGNED operates at this genome-scale resolution while jointly performing (i) response prediction and (ii) iterative refinement of the entire signed GRN.

**6. ALIGNED consistently outperforms SOTA across datasets** (Response to reviewer nsYx - Q7/YBGA - Q2). Following reviewer requests, we additionally compared a latest preprint SOTA, STATE, under the evaluation protocol compatible with our setting. Results show that **ALIGNED produces predictions better grounded in biological foundation than STATE, GEARS, scGPT, and scFoundation on multiple benchmark datasets.**

---

### Author Response · Authors · 2025-12-01
**General  Response for AC and Reviewers**

Dear AC and reviewers,

We sincerely thank AC and all reviewers for their careful evaluations and thoughtful feedback. We are grateful for their recognition of our work’s contributions toward **jointly optimizing neural predictors and symbolic gene-regulatory knowledge**. We also appreciate the acknowledgement of the **timeliness of this research direction**, the **originality of integrating biological prior knowledge with data-driven learning**, and the **potential impact of improving consistency** between genetic perturbation-response predictions and Gene Regulatory Networks (GRNs).

Across the reviews, we are encouraged by the positive remarks regarding:

- **Clear motivation and relevance** of resolving inconsistencies between curated GRNs and perturbation data, and the value of an adaptive neuro-symbolic alignment framework (as noted by reviewer Zesh, 6cxn and YBGA).
- **Conceptual novelty** of the bi-directional alignment/refinement loop compared to prior work (as noted by reviewer 6cxn and YBGA) and abductive reasoning with gradient-based updates (highlighted by reviewer Zesh).
- **Empirical promise** in improving both predictive performance and agreement with mechanistic knowledge (highlighted by reviewer Zesh and 6cxn).

We also greatly appreciate the reviewers’ suggestions for clarification and additional analysis. **In the revised manuscript, we have:**

- **Clarified assumptions and theory:** We made explicit the assumptions on neural and symbolic components, formalized the alignment and refinement operators.
- **Expanded ablations and cross-references:** We added ablations of individual components, additional adaptor/KB ablations, and new evaluations against external knowledge sources (e.g., PPI-based KBs).
- **Extended baselines:** We included comparisons with a preprint state-of-the-art method STATE in the manuscript and a recent mechanistic GRN-based perturbation simulator in our responses, alongside GEARS, scGPT, and scFoundation, under common benchmarking protocols.
- **Detailed hyperparameters and implementation:** We provided clearer descriptions of architectures, gene sets, hyperparameter selections (including $\theta$, $k$, $t$ and $\boldsymbol{w}$), and data splits, as well as additional performance statistics (e.g., precision/recall, training vs. held-out data).
- **Further clarifications:** We thank Reviewer nsYx for highlighting several points that were not sufficiently specified in the initial submission. We have substantially clarified these aspects in the revision and added the requested technical details, evaluations, and justifications.

All changes are highlighted in blue in the revised manuscript for ease of reference. We thank the reviewers again for their insightful comments, which have substantially improved the clarity, completeness, and empirical rigor of our work.

Below, we address each reviewer’s concerns in detail.

---

### Public Comment · ~Yuanfang_Xiang1 · 2026-03-04
**Response for Meta Review Comments**

We sincerely thank the ACs for meta review evaluations and for pointing out insufficient clarifications regarding the reviewer comments.

In the camera ready submission, we have further explained in Section 2.1 that ternary output is a commonly-used representation of gene expression profiles in downstream biological applications. We have added clarifications in Section 3.1 (page 3) and Section 4.1 (page 7) that our consistency metric evaluates the model’s generalization performance while remaining grounded in prior knowledge, and is not circular.

---

### Meta-Review · Area_Chair_Nvk3 · 2026-01-04

**Summary:**

Reviewers shared common concerns: the metric seems circular, continuous magnitudes are missing, training is costly, and wet-lab validation is absent.

**Reviewer Concerns:**

The rebuttal added convergence proofs, component ablations, comparisons with STATE, and computational-cost details, but the core worries that "the knowledge-consistency metric appears circular" and "ternary-only outputs may be insufficiently practical" were still left unresolved by two reviewers.

**Reviewer Scores:**

A few reviewers might raise their initial scores, but some concerns remain largely unresolved.

---

### Decision · Program_Chairs · 2026-01-26

Accept (Poster)